# Stimulus-Evoked Brain Signals for Parkinson’s Detection: A Comprehensive Benchmark Performance Analysis on Cross-Stimulation and Channel-Wise Experiments

**DOI:** 10.3390/bioengineering12111185

**Published:** 2025-10-30

**Authors:** Krishna Patel, Rajendra Gad, Marissa Lourdes de Ataide, Narayan Vetrekar, Teresa Ferreira, Raghavendra Ramachandra

**Affiliations:** 1School of Physical and Applied Sciences, Goa University, Taleigao Plateau 403206, Goa, India; elect.krishna@unigoa.ac.in (K.P.); rsgad@unigoa.ac.in (R.G.); marissaataide@unigoa.ac.in (M.L.d.A.); vetrekarnarayan@unigoa.ac.in (N.V.); 2Neurology Department, Goa Medical College and Hospital, Bambolim 403202, Goa, India; teresacf1980@gmail.com; 3Department of Information Security and Communication Technology, Norwegian University of Science and Technology (NTNU), 7491 Gjøvik, Norway

**Keywords:** Electroencephalogram (EEG), cross-stimulation, single channel analysis, Parkinson’s Disease (PD), classification

## Abstract

Parkinson’s disease (PD) is a progressive neurodegenerative disorder that affects both motor and cognitive functions, often resulting in misdiagnosis during its early stages. The condition severely impacts daily living, diminishing an individual’s ability to work and carry out routine tasks independently. Consequently, the development of automated methods for reliable PD detection has gained growing research interest. Among the available approaches, Electroencephalography (EEG) has emerged as a promising non-invasive and cost-effective tool. Nevertheless, most existing studies have predominantly focused on resting-state EEG, which constrains the generalizability and robustness of the proposed detection models. This study introduces a cross-stimulation evaluation framework to assess its impact on Parkinson’s disease detection algorithms and conducts channel-wise analysis to identify the most discriminative brain regions for accurate diagnosis. To support this research, we present the newly introduced Parkinson’s disease EEG (ParEEG) database, comprising 203,520 EEG samples from 60 subjects recorded based on Resting-State Visual Evoked Potential (RSVEP) and Steady-State Visually Evoked Potential (SSVEP) stimuli. In this study, we evaluate the performance of individual EEG channels using two handcrafted and two deep learning-based methods, employing a 10-fold cross-validation strategy, to ensure statistical reliability and establish benchmark results. Experimental results show that CRC and LSTM consistently achieved high accuracies (95–100%) with low variability (standard deviation < 2%). The analysis indicates that EEG channels in the frontal, fronto-central, and central–parietal regions consistently yield higher classification accuracy in Parkinson’s disease detection. Our findings offer valuable insights into channel-specific neural alterations for better interpretability in PD, and the cross-stimulation evaluation enhances the generalizability of EEG-based PD detection for practical diagnostic purposes.

## 1. Introduction

Parkinson’s disease (PD) is the second most common neurodegenerative disorder affecting approximately 2–3% of the global population with slightly higher prevalence in men than women [1], and this gap may vary by geographic/ethnic population [2]. PD is a complex, multifactorial syndrome characterized primarily by progressive motor impairments such as bradykinesia, tremor, rigidity, and postural instability [3,4]. These motor deficits arise largely from dopaminergic neuronal loss in the substantia nigra pars compacta, leading to dopamine depletion in the striatum. However, dopamine dysfunction also contributes to non-motor impairments [5], including deficits in executive function, working memory, attention, and visuospatial processing [6]. Importantly, PD pathology extends beyond dopaminergic pathways, as cholinergic, serotonergic, and noradrenergic systems are also affected, contributing to the diverse clinical manifestations of the disease [7]. Clinical diagnosis during early stages is often challenging, with considerable risk of misdiagnosis [8]. Advanced imaging techniques such as SPECT or cardiac MIBG can aid diagnosis but remain costly and less accessible. Alternatively, Electroencephalography (EEG) has growing interest as a low-cost, non-invasive neurophysiological tool for early PD detection. While EEG offers high temporal resolution and affordability, it may be influenced by artifacts arising from involuntary motor symptoms and the stress of EEG cap placement, which must be carefully considered in both research and clinical applications.

From the above noting, Electroencephalography (EEG) offers a non-invasive, affordable, and widely available means of capturing brain activity with high temporal resolution. While traditionally used in epilepsy and Alzheimer’s detection, recent studies have demonstrated its potential in identifying PD-related neural abnormalities [9,10]. However, EEG analysis is challenging due to its low signal-to-noise ratio and stochastic nature, motivating the integration of advanced computational methods, particularly machine learning and deep learning, for reliable PD detection [11,12]. Most of the PD detection studies rely on resting state EEG signals, recorded with eyes open or closed in a controlled, noise-free environment [13], ensuring that the EEG signals accurately reflect the brain’s resting state condition without external interference. However, beyond conventional resting-state analysis, researchers have begun exploring dynamic EEG responses under various stimulus-based conditions to better understand PD-related brain activity. Some studies have utilized cross-image presentations [14,15] and photic stimulation [16], where subjects are exposed to light flashes at different frequencies to elicit brain responses. Other investigations have examined reinforcement learning tasks [17], cognitive tasks [18], and hearing tasks [19], using voice recordings [20] and visual stimuli with foreground flickering [21] to analyze PD-related neural mechanisms. Additionally, emotional stimuli have been employed to study EEG responses to different emotional states such as happiness, fear, anger, disguised, sadness, surprise, and meditation [22,23]. Importantly, Parkinson’s disease is also associated with visual processing impairments, including reduced contrast sensitivity, abnormal eye movements, and delayed cortical responses to visual stimuli. A meta-analysis of visual evoked potential (VEP) studies reported significantly prolonged P100 latencies in PD patients compared to healthy controls, highlighting visual pathway dysfunction in the disease [24]. These findings provide strong motivation for using visually evoked paradigms, such as RSVEP and SSVEP, for PD detection.

Despite these recent advances, existing studies on Parkinson’s disease (PD) detection predominantly focus on analyzing behavioral and neural responses under specific stimulation or single stimulation conditions. Most of these studies are confined to intra-stimulation (within-stimulation) experiments, wherein classification models are trained and tested on EEG data obtained under the same type of stimulus. In contrast, this study explores the generalizability of PD detection algorithms through cross-stimulation experiments, utilizing EEG signals recorded under both Resting-State Visual Evoked Potential (RSVEP) and Steady-State Visually Evoked Potential (SSVEP) paradigms. This approach aims to evaluate the robustness and adaptability of PD detection methods across varying stimulus conditions.

Recent studies have also explored minimal-lead and single-channel EEG approaches for Parkinson’s detection. For instance, a recent work demonstrated that a single-channel forehead sensor combined with auditory task paradigms could differentiate PD patients and even predict F-DOPA PET outcomes, highlighting the feasibility of single-channel PD detection [25]. Similarly, several channel-selection and regional analysis studies have reported that a subset of frontal and central electrodes provides high discriminative power, suggesting that not all EEG channels are equally informative [26]. However, these approaches are either task-specific or limited to resting-state conditions. In contrast, our work provides a systematic channel-wise benchmarking under visually evoked paradigms (RSVEP and SSVEP), along with a cross-stimulation framework to test generalizability across conditions. This method enables the identification of the most discriminative individual channels for PD detection. Such a targeted approach offers two key advantages: (a) it highlights channels with the highest classification relevance, and (b) it reduces computational complexity by limiting analysis to the most informative channels. Moreover, this analysis facilitates the examination of region-specific brain responses to different stimulation paradigms, contributing to a more localized interpretation of Parkinson disease-related neural dysfunction.

To facilitate this research, we introduce a newly constructed Parkinson’s Disease EEG (ParEEG) database, comprising 203,520 EEG samples collected from 60 subjects (30 subjects belong to Healthy control and 30 subjects belong to Parkinson’s disease). The significance of the ParEEG database lies in recordings of brain wave signals from different emotional states, generated using Resting-State Visual Evoked Potential (RSVEP) and Steady-State Visual Evoked Potential (SSVEP) stimuli. To the best of our knowledge, cross-stimuli evaluation of PD detection using REVEP and SSVEP has not been extensively explored in prior research. In this study, we investigate extensive experimental analysis based on the average classification accuracy achieved by our PD detection algorithms, highlighting the effectiveness of single-channel cross-stimulation analysis. This approach demonstrates the potential for improved generalizability and diagnostic performance in PD detection across varying stimulus conditions. The primary contributions of this research include the following:Investigate a cross-stimulation evaluation framework to assess the generalizability and robustness of Parkinson’s disease detection algorithms across varying stimulus conditions, addressing a key shortcoming of prior work that primarily relies on intra-stimulus (within-stimulus) evaluations.Conduct a channel-wise performance analysis, evaluating classification accuracy at individual EEG channels to identify the most discriminative brain regions for PD detection across different stimulus conditions.Introduce the newly constructed ParEEG database, comprising 203,520 EEG samples from 60 subjects (30 healthy controls and 30 individuals with Parkinson’s disease), capturing EEG responses to diverse emotional states induced by Resting-State Visual Evoked Potential (RSVEP) and Steady-State Visually Evoked Potential (SSVEP) stimuli. The ParEEG dataset will be made publicly available for research purposes to support reproducible research.Present a comprehensive experimental analysis of PD detection algorithms within a cross-stimulation evaluation framework, benchmarking classification accuracy at the individual EEG channel level. The evaluation includes two handcrafted feature-based methods and two deep learning-based approaches, enabling an in-depth comparative assessment in handling variability across stimulus conditions.

The subsequent sections of this paper are structured as follows: Section 2 provides a comprehensive overview of the Parkinson’s EEG database and outlines the methodology employed for PD detection. Section 3 presents detailed experimental benchmark results based on intra-stimuli and cross-stimulation evaluation for individual channels, and Section 4 presents a discussion about the study and its findings, while Section 5 concludes the study by summarizing the key findings and insights, and outlines directions for future work.

## 2. Materials and Methods

### 2.1. Database Description

EEG data for this study were acquired using two distinct Data Collection Protocols, each incorporating audio–visual stimuli to assess neural responses. The sequences of stimuli used in these protocols are depicted in Figure 1. A Brain Product R-Net cap, equipped with 32 passive electrodes, along with a ground and reference electrode, was used to capture EEG signals at a sampling rate of 250 Hz, which is frequently adopted in research studies relevant to PD detection [27,28].

The dataset, referred to as ParEEG, consists of recordings from 60 volunteers, including 30 PD patients and 30 healthy control subjects. The age of participants ranged from 40 to 80 years. In the PD group, there were 20 males and 10 females, while the healthy control group consisted of 15 males and 15 females. The healthy control group was carefully age-matched with the PD group to minimize the confounding influence of age-related EEG differences. None of the healthy controls reported or exhibited any neurological, psychiatric, or major systemic disorders, and their health status was confirmed through a structured questionnaire and neurologist-supervised screening. All Parkinson’s disease patients were recruited from Goa Medical College and Hospital, Goa, India, under the supervision of a medical specialist. Their diagnosis was clinically established by a neurologist using the UKPDS Brain Bank Criteria, ensuring a standardized and reliable evaluation process. Additionally, the impairment levels of PD patients were documented, with approximately 70% exhibiting minimal and 30% exhibiting moderate impairment, providing a clear characterization of disease severity within the cohort. Recordings were performed with participants instructed to avoid their regular medication on the day of EEG acquisition to minimize pharmacological effects. The visual and emotional stimuli used in the experiments were obtained from publicly available sources (https://shorturl.at/5a1Cv, (accessed on 24 July 2025)). The selection and sequencing of these stimuli were performed and carried out under the supervision of an expert neurologist to ensure clinical relevance and appropriateness. To ensure data reliability, all EEG recordings were conducted in a controlled environment, with subjects seated comfortably to minimize movements and external disturbances. Visual stimuli were presented on a laptop screen positioned 76 cm from the participants, who were instructed to maintain a relaxed posture and limit eye blinking to reduce signal artifacts during EEG signal recordings. Further details of each Data Collection Protocol are outlined in the following subsections.

#### 2.1.1. Data Collection Protocol 1 (DCP 1)

Data Collection Protocol 1 is designed to investigate the neural mechanisms underlying resting-state activity and visually evoked responses, particularly in relation to cognitive and emotional processing. This protocol follows the Resting-State Visually Evoked Potential (RSVEP) paradigm [29]. Data collection under this session begins with a one-minute baseline EEG recording during eye closure, capturing spontaneous neural oscillations primarily dominated by alpha rhythms (8–12 Hz), which are known to be associated with a relaxed but wakeful state. One-minute baseline was chosen to maintain consistency across stimulus conditions and to minimize fatigue and discomfort in participants, especially those with PD. This is immediately followed by a ten-second eye-opening phase, during which the brain’s visual processing regions, particularly the primary visual cortex (V1) and associated occipital areas, become more engaged, leading to a suppression of alpha rhythms and an increase in beta and gamma activity. This cycle is repeated to ensure consistency in data acquisition [30].

To examine the influence of emotional stimuli on neural activity, participants are presented with one-minute video clips designed to elicit relaxation, amusement (comedy), and fear (horror). These stimuli are presented twice per category to ensure reliable neural responses. Emotional processing involves multiple brain regions, including the amygdala, prefrontal cortex, and limbic system, which interact with sensory processing areas to modulate EEG signals. For example, relaxing stimuli may enhance theta and alpha power, indicating a state of mental calmness, whereas horror stimuli may lead to increased beta and gamma activity, reflecting heightened alertness and emotional arousal. A schematic representation of the RSVEP sequence in Data Collection Protocol 1 is provided in Figure 1.

#### 2.1.2. Data Collection Protocol 2 (DCP 2)

Data Collection Protocol 2 is based on Steady-State Visually Evoked Potentials (SSVEPs), a paradigm for studying sustained visual attention, cognitive load, and brain–computer interface applications. SSVEPs are oscillatory brain responses elicited when a subject views a flickering stimulus at a specific frequency, primarily activating the occipital cortex (V1, V2, and V3) [31]. These responses are widely used to assess visual processing efficiency and cognitive workload, both of which are affected in neurodegenerative conditions such as Parkinson’s disease (PD).

In this Data Collection Protocol, participants complete four one-minute sessions involving flickering visual stimuli. During the initial sessions, they are presented with alphanumeric flickering sequences (characters 7, L, and T) oscillating within the 5–7.5 Hz frequency range. These visual stimuli are well-established for eliciting gamma-band activity in the visual cortex, thereby enabling the assessment of neural synchronization deficits in Parkinson’s disease (PD). The alphanumeric sequences are generated using MATLAB 2021a, with each character preceded by a one-second auditory cue. This cue is incorporated to promote multisensory integration and to engage higher-order cognitive networks, particularly the dorsolateral prefrontal cortex (DLPFC).

To examine the interaction between cognitive processing and emotional states, the flickering sequence is subsequently paired with video clips from the relaxation, comedy, and horror categories. These emotionally evocative videos serve as task-irrelevant distractions, providing a basis for analyzing attentional modulation in both Parkinson’s disease (PD) patients and healthy control groups. The effectiveness of attentional control, influenced by the parietal and prefrontal cortices, is crucial for cognitive function and is often impaired in PD patients. By measuring EEG responses under these varying conditions, this Data Collection Protocol provides insights into visual processing efficiency, attentional focus, and cognitive flexibility in individuals with PD and healthy controls. A schematic representation of Data Collection Protocol 2 is shown in Figure 1.

### 2.2. Pre-Processing

The EEG signals were initially preprocessed using a bandpass filter set with a cut-off frequency between 0.3 Hz and 100 Hz to retain relevant frequency components. This was then followed by a notch filter to eliminate 50 Hz power line interference. After preprocessing, the EEG recordings were segmented according to the different stimulus categories. Within each category, the continuous signals were further divided into 10 s intervals, resulting in EEG samples of size 32 * 2500, where 32 corresponds to the number of EEG channels and 2500 represents the number of data points, for given sampling rate.

In total, Data Collection Protocol 1 yielded 103,680 EEG samples, while Data Collection Protocol 2 produced 99,840 samples. A comprehensive breakdown of the sample distribution across different stimuli is provided in Table 1. To ensure clarity in referencing, unique notations were assigned to EEG recordings from different stimulation conditions as summarized in the same table. For example, the notation $horror_{p1}^{1}$ designates the second EEG sample recorded during the horror stimuli presentation condition in Data Collection Protocol 1 for the first instance. A block diagram illustrating the EEG-based PD detection process, starting from EEG data acquisition and pre-processing to classification, is presented in Figure 2.

### 2.3. Classification Methods

This section outlines the classification methodology used to differentiate between healthy controls and individuals diagnosed with Parkinson’s disease (PD). The analysis is performed at the level of individual EEG channels to evaluate their discriminative power. Specifically, a channel-wise approach is adopted, wherein the classifier is trained using data from a single EEG channel and subsequently tested on data with the same channel structure. Additionally, for cross-stimulation-based PD detection, the training and testing datasets consist of EEG signals recorded under different stimulation conditions. For instance, the training set may include EEG samples from the resting-state eye-closed condition ($rec_{p1}^{1}$), while the testing set uses EEG samples from the Horror clip ($horror_{p1}^{1}$) in the cross-stimulation evaluation.

Signals corresponding to individual stimuli are extracted independently for each Data Collection Protocol. For instance, in Data Collection Protocol 1, EEG recordings are obtained for eight different stimuli: $rec_{p1}^{1}$, $rec_{p1}^{2}$, $relax_{p1}^{1}$, $comedy_{p1}^{1}$, $horror_{p1}^{1}$, $relax_{P1}^{2}$, $comedy_{P1}^{2}$, and $horror_{P1}^{2}$, following the sequence detailed in Table 1.

To train the classification model, let N represent the number of training samples, where each sample consists of a single-channel EEG segment from either a healthy control or a Parkinson’s disease (PD) subject. The training dataset can be expressed as(1)$$T_{\mathrm{train}}=X_{i},Y_{i}\in R,\;\forall i=1,2,\ldots ,N$$
where $X_{i}$ represents an individual EEG sample from a specific electrode and $Y_{i}\in \left\{0,1\right\}$ denotes the class labels, with 0 for healthy controls and 1 for PD subjects.

For model evaluation, let M denote the number of testing samples. Each test sample undergoes classification based on the trained model, where the prediction function is given by(2)$$Y_{\mathrm{test}}\left(X\right)=\left\{\begin{array}{cc} 0, & \mathrm{if}\;X\in \mathrm{Healthy}\;\mathrm{Control}\\ 1, & \mathrm{if}\;X\in \mathrm{Parkinson}’s\;\mathrm{Disease} \end{array}\right.$$

This formulation ensures that the analysis is conducted at the individual channel level, allowing for an evaluation of which brain regions contribute most effectively to PD detection.

In this study, we employ four distinct algorithms for Parkinson’s disease detection, comprising two traditional handcrafted feature-based methods and two deep learning–based approaches. The selected methods include Support Vector Machine (SVM) [16], Collaborative Representation Classifier (CRC) [32], Long Short-Term Memory (LSTM) [23], and Convolutional Neural Network (CNN) [14]. A concise overview of each algorithm is provided in the following section.

#### 2.3.1. Support Vector Machine (SVM)

In this study, a linear SVM is utilized as a binary classifier [16] to differentiate between healthy individuals and those diagnosed with Parkinson’s disease (PD). The linear kernel is selected to efficiently determine the optimal hyperplane that separates the two classes in the feature space while maintaining computational efficiency.

The decision function of the SVM is mathematically represented as(3)$$g\left(c\right)=h\left(d\right)=\mathrm{sign}\{\alpha^{T}\cdot \psi \left(d\right)+\beta \}$$
where α denotes the weight vector that defines the orientation of the decision boundary, and ψ represents a transformation function that maps input features into a higher-dimensional space to enhance separability. The bias term β ensures optimal positioning of the decision boundary for maximum class distinction.

#### 2.3.2. Collaborative Representation Classifier (CRC)

CRC is an effective classification approach that applies collaborative representation principles combined with regularized least squares, making it well-suited for high-dimensional data such as EEG signals [32]. This method optimally reconstructs the test sample using a combination of training data, minimizing reconstruction error while ensuring generalization through regularization.

The CRC model is formulated as(4)$$d={\mathrm{argmin}}_{\sigma }{∥p-\delta_{tr}\sigma ∥}_{2}^{2}+\alpha {∥\sigma ∥}_{2}^{2}$$
where *p* represents the test sample requiring classification, δtr consists of the learned features from the training dataset, and σ is the coefficient vector that quantifies the contributions of different training samples in reconstructing the test sample. The regularization parameter α controls the trade-off between reconstruction accuracy and model complexity. In this study, $l_{2}-$regularization is employed, and an optimal performance is achieved with α=0.001 value.

#### 2.3.3. Long Short-Term Memory (LSTM)

For EEG signal classification, a bidirectional Long Short-Term Memory (BiLSTM) network is implemented to process sequential EEG data. The model’s input consists of single-channel EEG signals, represented as a time series of features. The network architecture begins with a sequence input layer, followed by a BiLSTM layer [23] with 70 hidden units. This bidirectional setup enables the model to extract both past and future temporal dependencies, which is crucial for EEG signal interpretation. The BiLSTM layer’s output is then fed into a fully connected layer, which maps the 70-dimensional feature space onto the two output classes. A softmax layer subsequently computes class probabilities, and the classification layer determines the final output.

The LSTM network is designed to process single-channel EEG signals extracted from each of the 32 electrodes independently. For each channel, the model is trained using the Stochastic Gradient Descent with Momentum (SGDM) optimizer, with an initial learning rate of 0.01, a maximum of 50 epochs, and a mini-batch size of 35, which provides a balance between convergence speed and generalization. To prevent exploding gradients during backpropagation through time, a gradient threshold of 1 is applied. The network architecture includes a sequence input layer with input size 1 (corresponding to single-channel data), followed by a bidirectional LSTM (BiLSTM) layer with 70 hidden units, which captures temporal dependencies in both forward and backward directions. The BiLSTM output is passed through a fully connected layer, followed by a softmax layer and a classification layer to perform binary classification between healthy and Parkinson’s EEG patterns.

#### 2.3.4. One-Dimensional Convolutional Neural Network (1D-CNN)

A lightweight 1D-CNN model is developed for classifying EEG signals into PD and healthy categories [14]. The architecture comprises two convolutional layers for feature extraction. The first layer employs 16 filters of size 5, while the second layer uses 8 filters. To preserve spatial dimensions, ‘causal’ padding is applied. Following each convolutional layer, a ReLU activation function introduces nonlinearity, and a normalization layer stabilizes training.

The convolutional operation is mathematically defined as(5)$$\mathrm{Output}\left(j\right)=\mathrm{ReLU}\left(\sum_{l=0}^{L-1}\theta \left(l\right)\cdot \mathrm{Signal}\left(j-l\right)+\gamma \right)$$
where θ(l) represents the convolutional filter weights, Signal(j) is the input EEG signal, and γ denotes the bias term.

After the convolutional layers, a global average pooling (GAP) layer is employed to condense the extracted feature maps into compact representations, effectively reducing model complexity while retaining the most salient information. The resulting feature vectors are then passed to a fully connected layer with two output nodes, corresponding to the binary classification task of distinguishing between healthy controls and Parkinson’s disease patients. A softmax layer generates class probabilities, and a classification layer determines the final predictions. The training is conducted using the SGDM optimizer, with an initial learning rate of 0.01, 50 epochs, and a mini-batch size of 35. To prevent exploding gradients, gradient clipping is implemented, ensuring stability during training.

### 2.4. Evaluation Method

This section presents the experimental evaluation method to obtain results using the newly developed ParEEG database, which consists of 203,520 EEG samples. The primary objective of these experiments is to evaluate Parkinson’s disease (PD) detection models, with a particular emphasis on channel-wise analysis and the effects of cross-stimulation evaluation on classification performance. Since EEG signals capture brain activity across multiple channels, each corresponding to electrical activity from different brain regions, a channel-wise approach provides valuable insights into the specific neural patterns associated with PD. Given that certain brain regions may exhibit more pronounced abnormalities than others, analyzing EEG channels individually helps identify the most discriminative ones, improving classification robustness and interpretability. This approach enables the assessment of regional variations in brain activity, enhances feature selection by focusing on highly informative channels, and offers the potential to optimize EEG setups by reducing the number of required electrodes. Notably, identifying the most relevant EEG channels can pave the way for single-channel EEG devices, making real-world implementation more feasible for clinical and wearable applications.

To achieve these objectives, we analyze the performance of four PD detection models: Support Vector Machine (SVM) [16], Collaborative Representation Classifier (CRC) [32], Long Short-Term Memory (LSTM) [23], and Convolutional Neural Network (CNN) [14]. Although the ParEEG dataset contains a large number of EEG segments (203,520), these are derived from a cohort of 60 subjects (30 PD and 30 healthy controls). To balance model complexity with the available cohort size and reduce the risk of overfitting, we employ lightweight deep learning architectures (CNN and LSTM). These models are configured with relatively few hidden layers, making them well-suited for small- to medium-sized datasets while still effectively capturing the spatial and temporal dynamics of EEG signals. To further reduce the risk of overfitting, classification was performed using 10-fold cross-validation at the subject level, ensuring that data from the same subject did not appear in both training and test sets. Each experiment is repeated 10 times with random splits, and average accuracy and standard deviation are reported to ensure reliable performance analysis and demonstrate statistical significance.

For a systematic evaluation, the ParEEG database is divided into training and testing subsets. Specifically, 50% of the subjects from both the healthy and PD groups, along with their respective EEG recordings, are allocated to the training set, while the remaining 50% constitute the testing set. The performance of PD detection models is examined under three distinct evaluation scenarios to incorporate channel-wise classification analysis.

Evaluation 1—Within-Stimulation: This serves as the baseline experiment, in which the training and testing EEG samples are derived from the same stimulus but are recorded at different time instances within a given Data Collection Protocol. The channel-wise accuracy distribution is examined to determine the most informative EEG channels contributing to reliable Parkinson’s disease detection.

Evaluation 2—Cross-Stimulation: The training and testing EEG samples belong to different stimuli within the same Data Collection Protocol (either Data Collection Protocol 1 or Data Collection Protocol 2). Channel-wise performance variation is examined to assess how evaluating different stimuli influence classification accuracy across individual EEG channel.

Evaluation 3—Cross-Stimulation: Training samples are taken from a specific stimulus (e.g., $rec_{p1}^{1}$) in Data Collection Protocol 1, while testing samples belong to a different stimulus (e.g., $relax_{p2}^{1}$) in Data Collection Protocol 2. This evaluation provides insight into how channel-wise PD detection generalizes across protocols with different stimulus conditions.

In all three evaluation scenarios, EEG samples are randomly selected in a non-overlapping manner, ensuring unbiased classification. The following subsections provide a detailed discussion of channel-wise results, highlighting key observations on the discriminative power of individual EEG channels in PD detection.

## 3. Results

This section presents the detailed experimental results based on the evaluation protocol described in Section 2.4. Further, detailed statistical results obtained for Evaluation 1, 2, and 3 are reported in the following subsections.

### 3.1. Evaluation 1

Table 2 and Table 3 demonstrate the classification accuracy obtained for Data Collection protocol (DCP) 1 and 2, respectively. Figure 3 and Figure 4 present the channel-wise performance comparison across four algorithms and illustrate the overall quantitative performance of the best-performing machine learning and deep learning algorithms respectively for DCP 1. Figure 5 and Figure 6 depict the corresponding results for DCP 2. Overall, an outstanding analysis is observed across all the algorithms, with no PD detection algorithm being exceptionally superior over other algorithms. While average classification accuracy varies across individual EEG channels, Table 2 and Table 3 reveal a remarkable 100% accuracy in Evaluation-1 for both DCP 1 and DCP 2 across several channels.

#### 3.1.1. Observations Related to Evaluation 1 Based on DCP 1

The key observations from evaluation 1 for EEG signals acquired under Data Collection 1 are summarized as follows:The CRC and LSTM algorithms demonstrated exceptional performance, attributable to the robustness of CRC in handling EEG signal variability and the capacity of LSTMs to capture long-term temporal dependencies in Parkinson’s EEG data. CNN also showed consistently strong performance, whereas SVM yielded comparatively lower accuracy, likely due to its reliance on manually extracted features that may not adequately represent the complex, nonlinear characteristics of EEG signals.Comparing the classification accuracy across different stimuli in DCP 1, the horror and comedy stimuli yielded comparatively better performance across most algorithms, suggesting that these stimuli may evoke stronger neural resonances, thereby enabling the models to more effectively differentiate Parkinson’s-affected EEG patterns from healthy ones.The best-performing EEG channels across all algorithms include frontal (Fp1, F9, F7), fronto-central (Fc5, Fc1, Fc2), central–parietal (Cp2), and parietal (P8), achieving an average classification accuracy ranging from 80% to 95%. This could be attributed to Parkinson’s disease being associated with widespread alterations in EEG spectral power and functional connectivity, particularly affecting the frontal and parietal regions [33,34]. In the eye-closed resting state, healthy individuals typically exhibit dominant alpha rhythms in posterior regions, which are often reduced or disrupted in PD patients [35]. Such disruptions manifest as altered activity patterns in parietal and central–parietal channels. Furthermore, during relaxed wakefulness, frontal and fronto-central regions often exhibit significant changes in EEG power and coherence in individuals with PD [36], contributing to distinguishable patterns that can support effective classification. These findings suggest that channel-wise EEG analysis is a valuable approach for identifying informative features and optimizing electrode selection in the PD detection system.

#### 3.1.2. Observations Related to Evaluation 1 Based on DCP 2

The key observations from evaluation 1 for EEG signals acquired under Data Collection 1 are summarized as follows:In DCP-2, the highest classification accuracy of 100% was achieved with the CRC and LSTM classification methods. Across all algorithms, classification accuracy was highest for the $L7T_{p2}^{1}$ Vs $L7T_{p2}^{2}$ evaluation, while it was relatively lower in $L7T_{p2}^{1}$ Vs $L7T_{p2}^{3}$ and $L7T_{p2}^{1}$ Vs $L7T_{p2}^{4}$ evaluations. This suggests that the L7T flickering pattern induces strong and consistent neural responses, particularly in short-term sequential comparisons, thereby facilitating more accurate differentiation of PD signals from healthy controls.The best-performing channels (F7, F9, Fc1, C3, P4, Cp2, Fc2, and Fp2) demonstrated high classification accuracy, spanning frontal, fronto-central, central, and parietal scalp regions. Their consistent performance highlights strong discriminative potential in Parkinson’s disease detection. The L7T flickering pattern, designed to elicit steady-state visual responses, may enhance neural activity effectively captured by fronto-central and parietal channels. The resulting stimulus-induced signal variations across these regions likely contribute to the enhanced classification performance observed under visual stimulation.

### 3.2. Evaluation 2

In the Within-Stimulation Evaluation, models are tested on the same stimulus recorded at different time points to capture intra-stimulus variations. However, this approach does not adequately assess the generalizability of algorithms across varying stimulus conditions. Evaluation 2 addresses this limitation by training and testing the PD detection models on different stimuli within the same Data Collection Protocol, thereby introducing EEG variability associated with distinct cognitive and emotional responses. This makes classification more challenging, providing a more rigorous test of model robustness under diverse neural conditions.

Table 4 and Table 5 demonstrate the classification accuracy obtained for DCP 1 and -2, respectively. Figure 7 and Figure 8 present the channel-wise performance comparison across four algorithms and illustrate the overall quantitative performance of the best-performing machine learning and deep learning algorithms respectively for DCP 1. Figure 9 and Figure 10 depict the corresponding results for DCP 2. Despite this increased complexity in this evaluation, the classification performance remains strong, demonstrating the robustness of the algorithms in handling EEG signal variations.

#### 3.2.1. Observations Related to Evaluation 2 Based on DCP 1

The key observations from evaluation 2 for EEG signals acquired under Data Collection 1 are summarized as follows:An outstanding performance was observed with CRC and LSTM algorithms, attaining the highest average classification accuracy of 99–100%, indicating their superior ability to capture spatial and temporal dependencies in Parkinson’s EEG signals. In comparison, CNN also demonstrated strong performance in PD detection, while SVM yielded the lowest accuracy among the evaluated methods.In Parkinson’s patients, emotional processing and cognitive engagement involve frontal and limbic system regions, which are often affected by neurodegeneration. This may influence EEG patterns differently across Horror and Comedy stimuli compared to the Relax stimulus, resulting in better classification accuracy. Such differences could be attributed to stronger neural activation in response to emotionally and cognitively engaging stimuli, leading to clearer differentiation between Parkinson’s and healthy EEG patterns. In contrast, the Relax stimulus might evoke weaker cortical responses, making classification more challenging.The best-performing channels across all algorithms include F7, F9, Fc5, Fc2, C3, Cp2, P3, and Fp2. The frontal and fronto-central areas (F7, F9, Fc5, and Fc2) are linked to cognitive processing and attention, which are likely heightened during emotional stimuli such as Horror and Comedy, thereby contributing to outstanding classification accuracy.

#### 3.2.2. Observations Related to Evaluation 2 Based on DCP 2

The key observations from evaluation 2 for EEG signals acquired under Data Collection 2 are summarized as follows:Again, CRC and LSTM consistently outperform other models, reinforcing their robustness in classifying Parkinson’s EEG signals. Accuracy peaks in Relax v/s Horror, reaching 93–100% for CRC and 93–99% for LSTM, which suggests a strong neural contrast between these conditions.Higher classification accuracy for Relax v/s Horror indicates that the horror stimulus elicits stronger neural activity compared to Relax, making the EEG patterns more distinguishable.The best-performing channels are located in frontal (F7, F9), fronto-central (Fc1), central (C3), parietal (P4, Cp2), and occipital regions. These regions are linked to emotion processing, motor control, and sensory integration, all of which are affected in Parkinson’s, thereby explaining their contribution to high classification accuracy.

### 3.3. Evaluation 3

Table 6 summarize the classification accuracy obtained from cross-stimuli evaluation. Figure 11 presents the channel-wise performance comparison of all four algorithms, and Figure 12 illustrates the overall quantitative performance of the two handcrafted and two deep learning algorithms. The key observations from this evaluation are summarized as follows:

This evaluation introduces greater variability by training on $rec_{p1}^{1}$ from Protocol 1 and testing on independent instances such as $relax_{p2}^{1}$, $comedy_{p2}^{1}$, $horror_{p2}^{1}$, and $7LT_{p2}^{1}$ from Protocol 2. The variation in recording conditions across protocols enables assessment of the models’ generalizability under different stimulus and temporal settings. As expected, CRC and LSTM achieve the highest accuracy, reaffirming their ability to capture spatial and temporal EEG patterns. CNN maintains consistent performance, whereas SVM lags, highlighting its sensitivity to cross-stimulation evaluation.Among the test stimuli, $rec_{p1}^{1,2}$ vs. $horror_{p2}^{1}$ achieves the highest classification accuracy across all algorithms. This indicates that horror stimuli evoke distinct EEG responses that enhance PD classification relative to other conditions. The stronger emotional and cognitive engagement associated with horror may likely lead to more pronounced neural differences between Parkinson’s and healthy subjects, thereby improving classification performance.The most effective channels—F7, F9, Fc1, Fc2, Fc5, C4, Cp2, and P8—are primarily located in the frontal, fronto-central, and central regions, which are crucial for motor control, cognitive processing, and sensorimotor integration. These regions, often affected in Parkinson’s disease, consistently yield reliable classification performance across evaluations, even under varying stimulus and protocol conditions.

**Figure 11 bioengineering-12-01185-f011:**
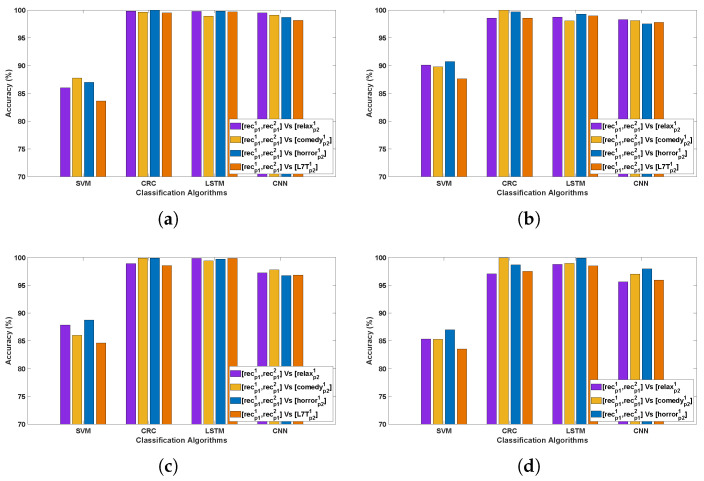
Illustrating the performance accuracy of the top four EEG channels across all four algorithms (Evaluation 3; only the four best-performing channels are presented for simplicity). (**a**) Channel 4 Accuracy Across Algorithms; (**b**) Channel 5 Accuracy Across Algorithms; (**c**) Channel 22 Accuracy Across Algorithms; (**d**) Channel 27 Accuracy Across Algorithms.

**Figure 12 bioengineering-12-01185-f012:**
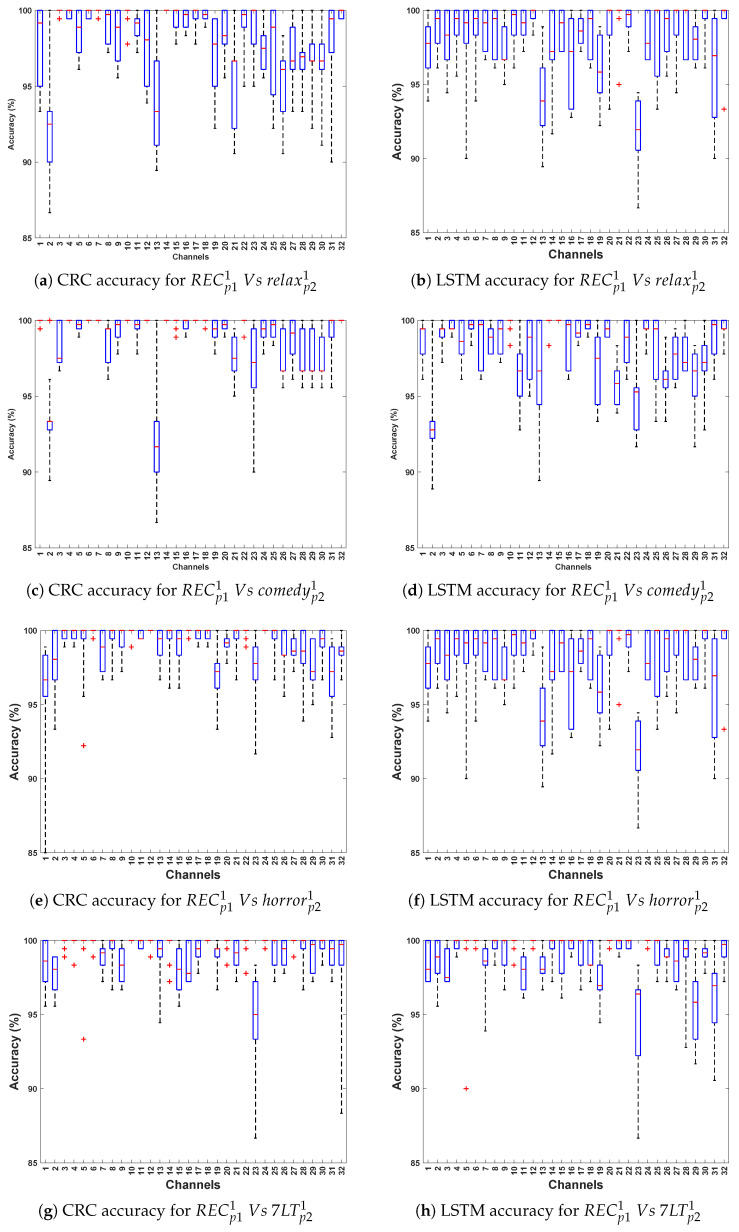
Illustrates the performance across 32 channels for PD detection (Evaluation 3; presenting the two best-performing algorithms: CRC and LSTM for simplicity).

**Table 6 bioengineering-12-01185-t006:** Average Classification accuracy of four PD detection algorithms for cross-stimulation Evaluation 3 using Data Collection Protocol 1 and Data Collection Protocol 2.

Ch	${\mathrm{rec}}_{p1}^{1,2}$ Vs ${\mathrm{relax}}_{p2}^{1}$				${\mathrm{rec}}_{p1}^{1,2}$ Vs ${\mathrm{comedy}}_{p2}^{1}$				${\mathrm{rec}}_{p1}^{1,2}$ Vs ${\mathrm{horror}}_{p2}^{1}$				${\mathrm{rec}}_{p1}^{1,2}$ Vs $7{\mathrm{LT}}_{p2}^{1}$			
	SVM	CRC	LSTM	CNN	SVM	CRC	LSTM	CNN	SVM	CRC	LSTM	CNN	SVM	CRC	LSTM	CNN
1	80.6 ± 4.5	97.7 ± 2.8	95.8 ± 3.8	98.4 ± 1.1	79.9 ± 2.5	97.8 ± 2.1	97.3 ± 1.9	97.1 ± 1.6	81.5 ± 4.7	99.9 ± 0.2	98.3 ± 1.6	98.7 ± 1.1	79.0 ± 3.7	98.8 ± 1.3	98.4 ± 1.2	96.5 ± 0.9
2	72.2 ± 3.3	92.4 ± 3.5	97.8 ± 2.2	95.1 ± 3.3	71.2 ± 3.1	96.8 ± 2.1	98.8 ± 1.5	97.5 ± 1.3	72.4 ± 3.2	93.4 ± 3.0	97.7 ± 1.2	93.3 ± 2.0	70.5 ± 3.5	93.2 ± 3.2	98.7 ± 1.4	97.7 ± 1.5
3	79.8 ± 3.8	99.9 ± 0.2	99.7 ± 0.4	98.3 ± 1.3	78.5 ± 4.6	99.1 ± 1.2	98.0 ± 1.9	98.7 ± 0.7	80.6 ± 4.5	98.3 ± 1.5	99.8 ± 0.4	98.5 ± 1.4	78.6 ± 4.2	99.2 ± 0.8	98.3 ± 1.2	97.9 ± 1.7
4	86.0 ± 4.2	99.8 ± 0.3	99.8 ± 0.4	99.5 ± 0.4	87.8 ± 3.6	99.6 ± 1.1	98.9 ± 1.4	99.1 ± 0.8	87.0 ± 4.0	100.0 ± 0.0	99.8 ± 0.5	98.7 ± 0.8	83.6 ± 3.1	99.5 ± 0.4	99.7 ± 0.4	98.2 ± 1.5
5	90.1 ± 3.3	98.6 ± 1.5	98.7 ± 2.5	98.3 ± 2.4	89.8 ± 2.8	100.0 ± 0.0	98.1 ± 2.9	98.1 ± 3.5	90.7 ± 3.1	99.7 ± 0.4	99.3 ± 2.0	97.5 ± 2.2	87.6 ± 3.2	98.6 ± 1.4	98.9 ± 3.0	97.8 ± 2.9
6	83.4 ± 3.4	99.8 ± 0.3	99.9 ± 0.2	98.0 ± 2.0	81.2 ± 3.0	98.3 ± 1.4	98.8 ± 1.8	97.9 ± 1.0	83.7 ± 4.1	100.0 ± 0.0	99.9 ± 0.3	98.6 ± 0.9	83.3 ± 4.1	99.6 ± 0.5	99.9 ± 0.2	98.6 ± 1.0
7	84.4 ± 3.8	99.9 ± 0.2	98.7 ± 1.2	98.2 ± 1.7	82.8 ± 3.7	98.9 ± 1.1	98.8 ± 1.3	97.9 ± 1.0	85.3 ± 3.9	100.0 ± 0.0	98.9 ± 0.9	96.7 ± 1.8	83.0 ± 3.3	98.8 ± 1.6	98.2 ± 1.9	97.9 ± 0.7
8	81.9 ± 3.3	99.1 ± 1.2	99.4 ± 1.0	95.7 ± 1.7	83.2 ± 3.8	99.6 ± 1.1	98.4 ± 1.6	96.6 ± 2.2	83.2 ± 3.4	98.6 ± 1.4	99.5 ± 1.0	96.4 ± 2.2	77.6 ± 3.1	98.8 ± 0.9	99.7 ± 0.5	97.2 ± 1.8
9	74.0 ± 3.1	98.3 ± 1.9	99.4 ± 1.0	99.0 ± 0.9	73.2 ± 2.8	100.0 ± 0.0	97.3 ± 1.6	98.7 ± 1.3	74.4 ± 3.6	99.3 ± 0.8	98.4 ± 1.2	98.7 ± 1.1	73.3 ± 2.7	98.8 ± 1.2	99.2 ± 1.1	98.3 ± 1.2
10	58.9 ± 3.6	99.7 ± 0.7	99.9 ± 0.3	97.9 ± 1.3	59.5 ± 3.6	98.7 ± 1.4	98.9 ± 1.4	97.9 ± 1.0	58.8 ± 2.9	100.0 ± 0.0	100.0 ± 0.0	97.3 ± 2.4	59.2 ± 4.8	99.8 ± 0.5	99.8 ± 0.5	98.1 ± 1.5
11	66.7 ± 4.3	98.9 ± 0.9	99.6 ± 0.3	96.1 ± 2.2	68.5 ± 5.8	99.4 ± 0.4	99.0 ± 0.9	95.8 ± 1.4	66.9 ± 4.5	99.4 ± 0.8	99.8 ± 0.3	96.1 ± 3.1	67.0 ± 4.1	96.6 ± 2.4	97.9 ± 1.1	94.3 ± 1.1
12	75.8 ± 4.3	97.6 ± 2.5	100.0 ± 0.0	96.7 ± 1.8	76.3 ± 4.0	99.3 ± 0.4	99.6 ± 0.7	95.3 ± 2.2	77.4 ± 5.5	100.0 ± 0.0	99.8 ± 0.4	95.4 ± 2.7	74.1 ± 2.9	98.1 ± 2.0	99.9 ± 0.2	96.5 ± 1.5
13	69.6 ± 2.6	94.0 ± 3.5	99.1 ± 1.2	96.2 ± 2.4	68.7 ± 4.1	93.0 ± 4.2	94.1 ± 2.8	93.3 ± 4.6	69.7 ± 2.9	92.3 ± 3.9	98.9 ± 1.7	88.7 ± 4.0	68.4 ± 3.2	96.3 ± 3.5	98.3 ± 1.0	95.1 ± 2.6
14	80.0 ± 4.4	100.0 ± 0.0	99.4 ± 1.2	98.1 ± 2.0	79.6 ± 5.1	98.3 ± 1.8	97.2 ± 3.0	96.6 ± 2.0	80.0 ± 4.4	100.0 ± 0.0	99.6 ± 0.9	97.7 ± 1.4	80.0 ± 4.4	99.8 ± 0.5	99.6 ± 0.8	97.4 ± 2.4
15	77.1 ± 5.2	99.4 ± 0.8	98.9 ± 1.4	98.4 ± 1.2	79.7 ± 5.1	99.8 ± 0.3	98.7 ± 1.3	97.9 ± 1.1	77.2 ± 5.4	99.8 ± 0.4	97.9 ± 1.4	98.3 ± 1.0	76.8 ± 5.1	100.0 ± 0.0	99.1 ± 1.3	98.7 ± 1.0
16	77.7 ± 5.5	99.4 ± 0.7	99.9 ± 0.2	96.7 ± 1.9	76.4 ± 4.3	98.1 ± 1.6	96.7 ± 2.8	95.5 ± 2.0	78.6 ± 5.4	99.7 ± 0.5	98.3 ± 1.2	96.1 ± 4.1	76.0 ± 5.4	98.8 ± 1.6	99.7 ± 0.5	96.3 ± 1.5
17	74.8 ± 4.1	99.7 ± 0.7	99.7 ± 0.4	98.6 ± 1.5	76.5 ± 4.2	99.4 ± 1.1	98.7 ± 0.9	97.3 ± 1.0	75.5 ± 4.8	100.0 ± 0.0	99.2 ± 0.7	98.4 ± 1.4	73.9 ± 4.3	99.3 ± 0.7	99.1 ± 1.2	98.3 ± 1.2
18	73.5 ± 4.2	99.7 ± 0.4	99.7 ± 0.5	96.7 ± 2.5	73.1 ± 5.2	99.9 ± 0.2	98.5 ± 1.6	97.4 ± 1.6	74.5 ± 4.9	99.9 ± 0.2	100.0 ± 0.0	97.4 ± 2.0	71.8 ± 4.4	99.7 ± 0.4	98.9 ± 1.0	97.7 ± 0.9
19	76.4 ± 9.6	97.1 ± 2.6	96.7 ± 1.9	96.2 ± 2.4	75.8 ± 8.0	99.7 ± 0.3	96.1 ± 2.0	98.2 ± 1.2	76.8 ± 9.3	99.3 ± 0.8	99.1 ± 0.9	96.1 ± 2.4	75.2 ± 8.3	96.9 ± 2.4	97.3 ± 1.6	96.1 ± 2.5
20	85.7 ± 2.5	98.4 ± 1.5	99.2 ± 0.6	96.6 ± 2.6	85.2 ± 2.8	99.6 ± 0.4	98.8 ± 2.0	95.7 ± 1.7	86.7 ± 3.1	99.6 ± 0.5	99.8 ± 0.5	96.4 ± 1.7	84.9 ± 2.2	99.4 ± 0.5	99.9 ± 0.2	96.2 ± 1.7
21	77.2 ± 8.1	95.2 ± 3.1	99.6 ± 1.0	98.4 ± 1.4	80.9 ± 7.5	100.0 ± 0.0	99.4 ± 1.5	98.1 ± 1.1	79.0 ± 8.8	97.6 ± 1.6	99.1 ± 1.0	97.9 ± 0.8	76.2 ± 7.3	95.8 ± 1.5	99.7 ± 0.4	96.6 ± 1.8
22	87.8 ± 2.7	98.9 ± 1.7	99.8 ± 0.4	97.2 ± 2.6	86.0 ± 3.7	99.9 ± 0.2	99.4 ± 0.8	97.8 ± 1.2	88.7 ± 3.8	99.9 ± 0.4	99.7 ± 0.7	96.7 ± 2.5	84.6 ± 3.7	98.6 ± 1.5	99.8 ± 0.3	96.8 ± 1.0
23	63.8 ± 2.9	98.8 ± 1.8	97.2 ± 2.4	97.8 ± 2.4	67.5 ± 4.2	93.4 ± 3.0	91.6 ± 2.3	94.9 ± 1.2	65.3 ± 3.3	96.7 ± 3.3	94.6 ± 3.2	98.5 ± 1.1	64.0 ± 2.6	95.1 ± 2.4	94.5 ± 3.5	95.8 ± 2.6
24	77.0 ± 4.6	97.3 ± 1.4	100.0 ± 0.0	98.8 ± 0.7	79.3 ± 4.4	99.6 ± 0.5	98.3 ± 1.5	98.9 ± 1.0	78.1 ± 4.7	99.3 ± 0.8	99.9 ± 0.2	98.6 ± 1.3	76.6 ± 4.1	99.4 ± 0.7	99.9 ± 0.2	98.8 ± 0.8
25	84.5 ± 4.4	97.5 ± 3.0	99.4 ± 1.0	96.2 ± 2.7	84.2 ± 4.8	99.6 ± 0.4	98.4 ± 2.5	98.4 ± 1.1	86.0 ± 3.8	99.4 ± 0.6	99.3 ± 1.1	97.3 ± 2.1	82.3 ± 4.0	98.2 ± 2.4	99.3 ± 1.0	96.3 ± 2.7
26	69.0 ± 3.5	95.4 ± 2.3	98.5 ± 1.3	97.7 ± 1.8	72.7 ± 6.0	97.8 ± 1.7	98.8 ± 1.5	97.4 ± 1.5	70.3 ± 4.4	97.6 ± 1.6	99.1 ± 0.9	97.5 ± 1.9	68.9 ± 3.3	96.2 ± 1.4	99.0 ± 0.7	97.3 ± 1.5
27	85.3 ± 1.9	97.1 ± 2.1	98.8 ± 0.9	95.6 ± 2.2	85.3 ± 2.3	99.9 ± 0.2	98.9 ± 1.8	97.0 ± 1.6	87.0 ± 1.9	98.7 ± 1.4	99.9 ± 0.3	97.9 ± 1.5	83.5 ± 1.9	97.5 ± 1.5	98.5 ± 1.3	95.9 ± 2.8
28	79.6 ± 3.4	96.6 ± 1.9	98.4 ± 1.8	97.6 ± 1.5	82.6 ± 3.9	98.1 ± 1.6	98.9 ± 1.5	97.3 ± 1.6	80.4 ± 3.7	97.7 ± 1.7	99.4 ± 1.0	97.2 ± 2.6	78.3 ± 3.4	97.9 ± 1.4	98.8 ± 2.1	98.3 ± 1.9
29	73.6 ± 3.4	96.6 ± 2.1	97.6 ± 1.4	97.3 ± 1.3	75.0 ± 5.9	97.9 ± 1.7	97.9 ± 1.3	97.4 ± 1.2	73.6 ± 4.0	97.7 ± 1.7	99.1 ± 1.1	96.6 ± 1.4	73.0 ± 3.6	96.0 ± 2.3	95.7 ± 2.4	96.4 ± 2.1
30	73.8 ± 4.1	96.6 ± 2.6	99.1 ± 1.0	97.8 ± 1.5	74.7 ± 2.8	98.8 ± 1.4	99.2 ± 1.3	97.2 ± 1.5	75.8 ± 4.6	97.6 ± 1.6	99.7 ± 0.5	95.7 ± 3.9	72.7 ± 3.2	97.3 ± 2.1	99.1 ± 0.7	98.7 ± 0.9
31	79.6 ± 2.7	97.9 ± 3.3	96.9 ± 2.0	98.0 ± 1.1	81.0 ± 5.1	100.0 ± 0.0	96.1 ± 3.4	98.4 ± 1.2	80.6 ± 3.3	99.2 ± 1.4	99.1 ± 1.1	97.7 ± 1.2	77.6 ± 2.7	98.9 ± 1.4	96.2 ± 2.5	97.1 ± 1.3
32	80.7 ± 4.6	99.8 ± 0.3	98.6 ± 0.9	98.9 ± 1.0	82.7 ± 4.1	99.7 ± 1.1	99.2 ± 2.0	98.7 ± 0.7	82.8 ± 4.0	100.0 ± 0.0	98.3 ± 3.4	98.1 ± 1.2	80.1 ± 4.4	99.4 ± 0.6	99.4 ± 0.8	98.8 ± 0.7

## 4. Discussion

EEG has emerged as a powerful non-invasive biomarker for Parkinson’s disease (PD) detection, with recent studies employing resting-state, visually evoked potentials, and task-induced paradigms to explore PD-related brain dysfunction [37]. However, beyond conventional resting-state analysis, researchers have increasingly explored dynamic EEG responses under diverse stimulus-based conditions to better capture PD-related neural changes [14,15,16,22,23]. In this context, our study introduces a novel hybrid stimulus-driven EEG Data Collection Protocol, which integrates two complementary paradigms: Resting-State Visually Evoked Potential (RSVEP) and Steady-State Visually Evoked Potential (SSVEP). Data Collection Protocol 1 (RSVEP-based) involves baseline recordings (eye-closed/open) and affective video stimuli (relaxation, comedy, and horror), while Data Collection Protocol 2 (SSVEP-based) features flickering alphanumeric sequences coupled with emotional video distractions. These Data Collections Protocols were designed to enrich the temporal and cognitive diversity of evoked EEG responses across 32 channels. By capturing both spontaneous and stimulus-driven neural activity, our approach provides a multidimensional view of PD-related brain alterations and offers a significant advancement over prior studies limited to either resting-state or single-frequency SSVEP paradigms.

To further enhance interpretability and clinical relevance, we performed a detailed single-channel analysis to evaluate the contribution of individual EEG electrodes to Parkinson’s disease (PD) detection. Each of the 32 channels was assessed independently across all experiments using both handcrafted and deep learning classifiers. This analysis enabled the identification of brain regions most strongly associated with PD-related abnormalities. Notably, channels located in the frontal (F7, F9), fronto-central (FC2), and central–parietal (CP2) regions consistently achieved the highest classification accuracies, often exceeding 95–100% in both CRC and LSTM classifier models across a variety of stimulus conditions and evaluation setups. These results are in strong agreement with prior studies [38,39], which have highlighted frontal and central regions as key sites of dysfunction in PD, particularly with respect to motor planning, execution, and cognitive control. Motor symptoms such as bradykinesia, rigidity, and tremor are strongly linked to disrupted beta-band activity and impaired fronto-central network dynamics, which are reflected in the high discriminative power of these regions [40]. In addition, non-motor features of PD, including cognitive impairment and visual processing deficits, are associated with altered oscillatory activity in frontal and parietal networks [41]. This aligns with our observation that stimulus-driven responses, particularly those involving visual evoked activity, reveal distinct differences between PD patients and healthy controls.

Channel-wise evaluation offers a more targeted and interpretable perspective; these benchmarking evaluation results can facilitate the design of more efficient and clinically viable EEG configurations, potentially reducing the number of electrodes required without compromising diagnostic performance. Building upon this channel-specific analysis, we presented an extensive three-tier evaluation framework to rigorously assess the robustness and generalizability of our classification models and presented the benchmark results. Traditional EEG-based PD detection studies often rely on intra-stimulus (within-stimulus) evaluation, where training and testing data are derived from the same type of stimulus, thereby limiting the ability to assess model performance under real-world variability [42,43,44]. In contrast, our framework progresses through three increasingly challenging evaluation protocols. The first level involves within-stimulus classification, where training and testing are performed on the same stimulus but at different time instances—serving as a baseline. The second level introduces cross-stimulus learning within the same Data Collection Protocol, training on one stimulus and testing on another (e.g., horror vs. comedy). The third and most rigorous level performs inter-protocol cross-stimulus evaluation, where models trained on Data Collection Protocol 1 are tested on different stimuli from Data Collection Protocol 2, simulating a realistic generalization challenge.

Across all three evaluations, our results consistently demonstrate the superior performance with CRC and LSTM classifiers, with accuracies frequently exceeding 95%. This multi-stage evaluation strategy allowed us to investigate not only the spatial relevance of individual EEG channels but also the temporal and contextual stability of PD-related neural features under diverse cognitive and emotional stimuli. Importantly, it highlights the potential of emotionally rich stimuli (such as horror and comedy) to enhance discriminability between PD and healthy controls, reinforcing the value of stimulus-based EEG paradigms. Overall, this robust framework goes beyond existing methods by accounting for both spatial (channel-specific) and contextual (stimulus-dependent) variability, offering a more reliable pathway toward the development of clinically deployable EEG-based PD diagnostic tools.

Although the present study demonstrates that a few central and frontal electrodes consistently yield high discriminative power for PD detection, translating these findings into a practical single-channel or reduced-channel diagnostic device presents challenges. First, EEG signals have inherently low signal-to-noise ratio, and reliance on a single electrode may increase susceptibility to artifacts (e.g., muscle, ocular, or environmental noise). Second, achieving consistent electrode placement is critical, as small deviations can alter recorded signals and compromise reproducibility. Third, extensive validation across larger, more diverse populations is required to establish the clinical reliability of such minimal configurations. Therefore, while our results suggest the feasibility of reduced-channel PD detection, further research and technological development are necessary before such systems can be deployed in clinical practice.

This study provides encouraging results for EEG-based PD detection, and several avenues can further extend its scope. While classification of PD versus healthy controls is primarily exploratory rather than a clinical diagnostic application, this study establishes a baseline proof-of-concept framework. First, the resting-state baseline duration was limited to 60 s to minimize fatigue and maintain consistency across conditions; however, longer recordings (e.g., 3–5 min) may capture additional stable resting-state features and should be explored in future work. Second, the EEG acquisition was performed at 250 Hz, which adequately covers PD-related changes in the delta–beta range, but higher sampling rates (≥512 Hz) could enable investigation of gamma-band activity and finer temporal dynamics. Although participants refrained from medication on the day of recording to minimise pharmacological effects, dopaminergic medications (e.g., carbidopa levodopa) may still influence beta-band EEG activity, which can be systematically examined with signal-level analyses in future research, future studies could incorporate longer washout periods to better isolate disease-specific neural signals. Similarly, future studies will incorporate detailed UPDRS motor subscores to examine EEG alterations across multiple symptom dimensions and consider the potential presence of atypical Parkinsonism (e.g., drug-induced and vascular), despite all patients being clinically diagnosed using UKPDS Brain Bank Criteria, with future work adopting the updated MDS Clinical Diagnostic Criteria to align with evolving standards and enhance comparability across studies.

Future work will focus on expanding the ParEEG database to include additional Parkinsonian syndromes such as PSP and MSA, systematically evaluating medication effects, incorporating UPDRS subscores, and performing longitudinal data collection to enhance clinical relevance. These steps aim to build upon the baseline results presented in this study.

Finally, future research could explore a broader range of stimuli, including motor tasks and cognitive challenges, as PD patients commonly exhibit bradykinesia, tremors, and cognitive impairments. Analyzing EEG responses to such tasks may provide deeper insights into disease-specific neural dysfunctions and further enhance detection accuracy.

## 5. Conclusions and Future Work

With the global rise in Parkinson’s disease prevalence, the need for accurate and early detection methods is receiving increasing attention. Electroencephalography (EEG) has emerged as a valuable tool for the detection and monitoring of Parkinson’s disease (PD). This study investigated a cross-stimulation evaluation framework for Parkinson’s disease (PD) detection using EEG signals and examined the impact of different brain regions and stimulus conditions on classification performance. To achieve this, analysis was performed on two handcrafted classifiers, including Collaborative Representation-based Classification (CRC) and Support Vector Machine (SVM), and two deep learning models—Long Short-Term Memory (LSTM) and Convolutional Neural Network (CNN). The evaluations were performed on the Parkinson’s disease EEG (ParEEG) database, comprising 203,520 EEG samples from 60 subjects, including 30 unique individuals belongs to healthy control and 30 individuals belongs to parkinson disease. The experimental results demonstrated that CRC and LSTM consistently outperformed other classification models across different evaluation protocols, highlighting their robustness in detecting PD-affected EEG signals.

Notably, channels located in the frontal (F7, F9), fronto-central (FC2), and central (CP2) regions consistently achieved the highest classification accuracies—often exceeding 95–100% in both CRC and LSTM classifier models across a variety of stimulus conditions and evaluation setups. These findings suggest that future research may not require full-brain EEG recordings, as focusing on these regions could be sufficient for accurate PD classification. Additionally, multi-pattern evaluation through within-stimulus and cross-stimulus classification revealed that emotional stimuli influenced classification performance, with Horror and Comedy stimuli yielding the highest accuracy. The high classification accuracy, particularly for emotionally engaging stimuli, underscores the potential of stimulus-based EEG analysis for PD detection. In the current work, the stimuli were exclusively visual, focusing on RSVEP, SSVEP, and emotional video paradigms. Future studies will incorporate motor tasks (e.g., finger tapping and gait-related EEG paradigms) and cognitive challenges (e.g., working memory tasks), which may provide complementary insights into PD-related neural dysfunction and improve the sensitivity of EEG-based biomarkers.

Given these promising outcomes, the proposed cross-stimulation evaluation framework benchmarked across individual channels holds significant potential as a reliable tool for aiding clinicians in PD diagnosis. The strong classification performance across multiple evaluation settings further highlights the feasibility of EEG-based biomarkers and deep learning techniques in advancing PD detection.

## Figures and Tables

**Figure 1 bioengineering-12-01185-f001:**
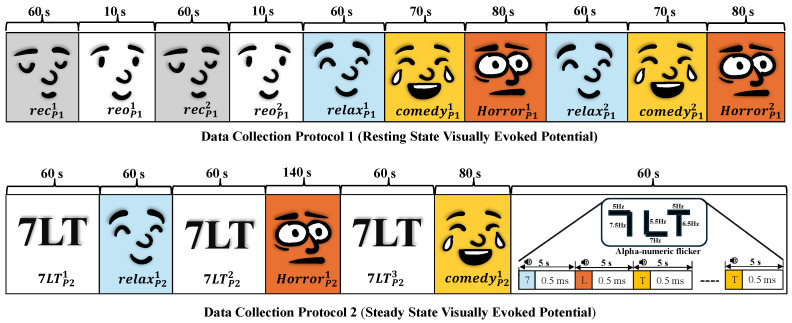
Temporal representation of audio–visual stimuli for EEG acquisition across Data Collection Protocol 1 (DCP 1) and Data Collection Protocol 2 (DCP 2).

**Figure 2 bioengineering-12-01185-f002:**
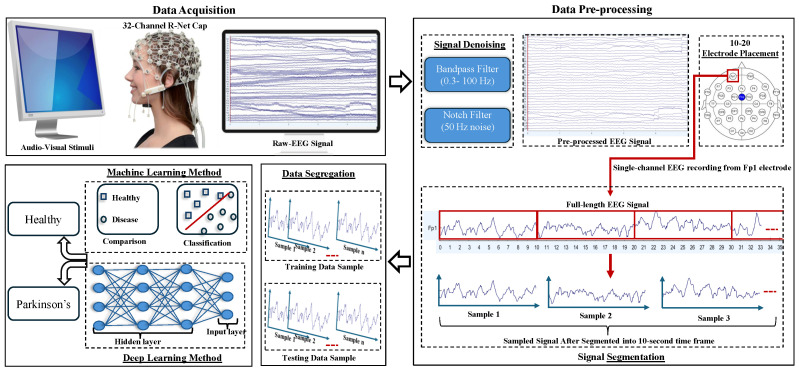
Block diagram illustrates the process of Parkinson’s disease detection using brain wave signals.

**Figure 3 bioengineering-12-01185-f003:**
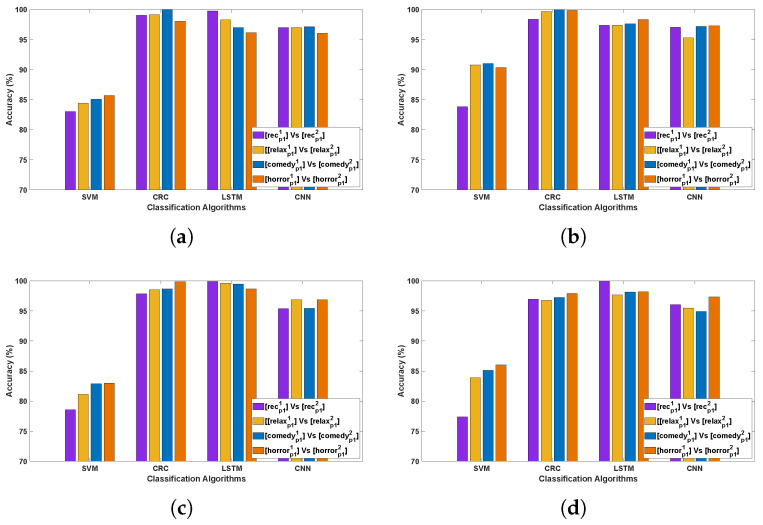
Illustrating the performance accuracy of the top four EEG channels across all four algorithms for DCP 1 (Evaluation 1; only the four best-performing channels are presented for simplicity). (**a**) Channel 4 Accuracy Across Algorithms; (**b**) Channel 5 Accuracy Across Algorithms; (**c**) Channel 22 Accuracy Across Algorithms; (**d**) Channel 27 Accuracy Across Algorithms.

**Figure 4 bioengineering-12-01185-f004:**
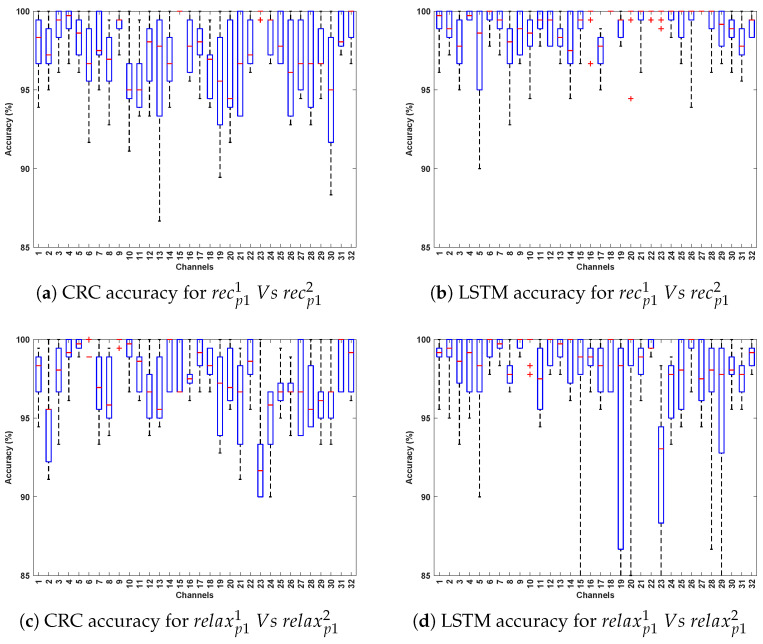
Illustrates the performance across 32 channels for PD detection for DCP 1 (Evaluation 1; presenting the two best-performing algorithms: CRC and LSTM for simplicity).

**Figure 5 bioengineering-12-01185-f005:**
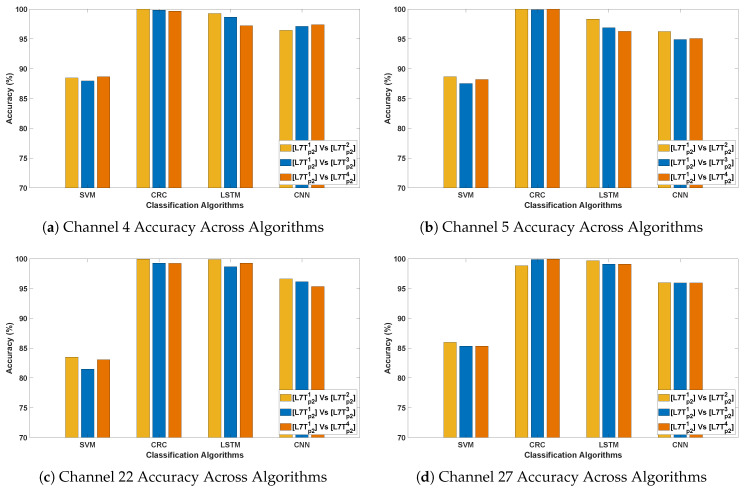
Illustrating the performance accuracy of the top four EEG channels across all four algorithms for DCP 2 (Evaluation 1; only the four best-performing channels are presented for simplicity).

**Figure 6 bioengineering-12-01185-f006:**
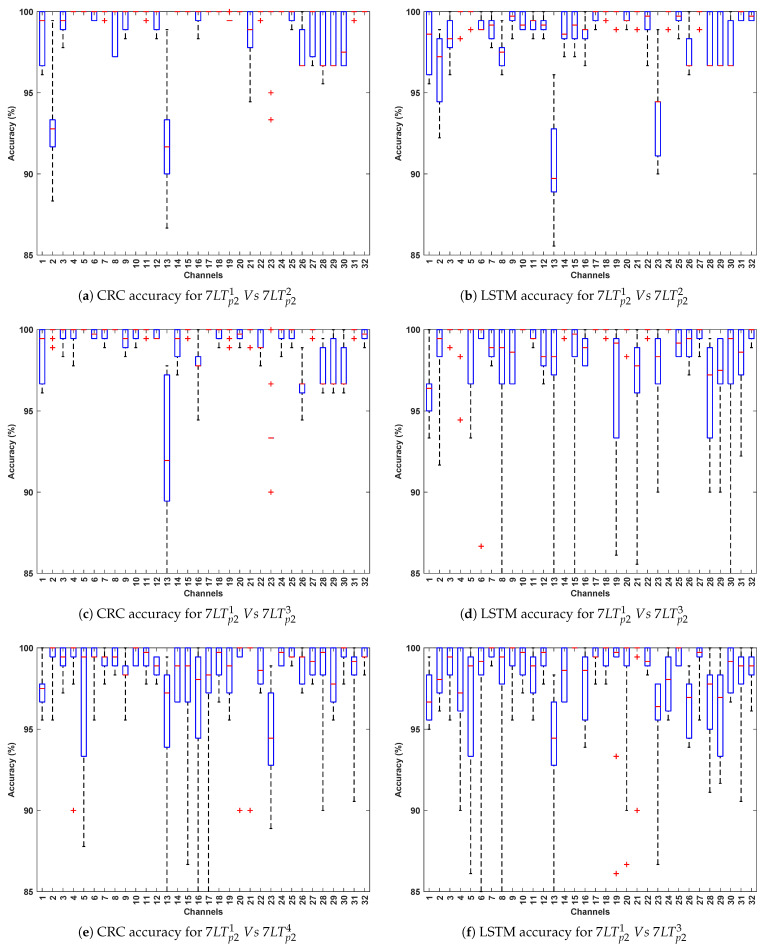
Illustrates the performance across 32 channels for PD detection for DCP 2 (Evaluation 1; presenting the two best-performing algorithms: CRC and LSTM for simplicity).

**Figure 7 bioengineering-12-01185-f007:**
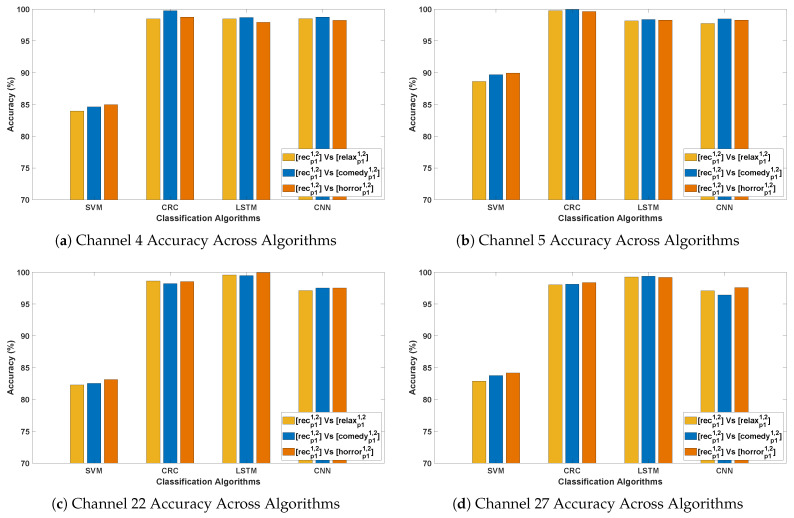
Illustrating the performance accuracy of the top four EEG channels across all four algorithms for DCP 1 (Evaluation 2; only the four best-performing channels are presented for simplicity).

**Figure 8 bioengineering-12-01185-f008:**
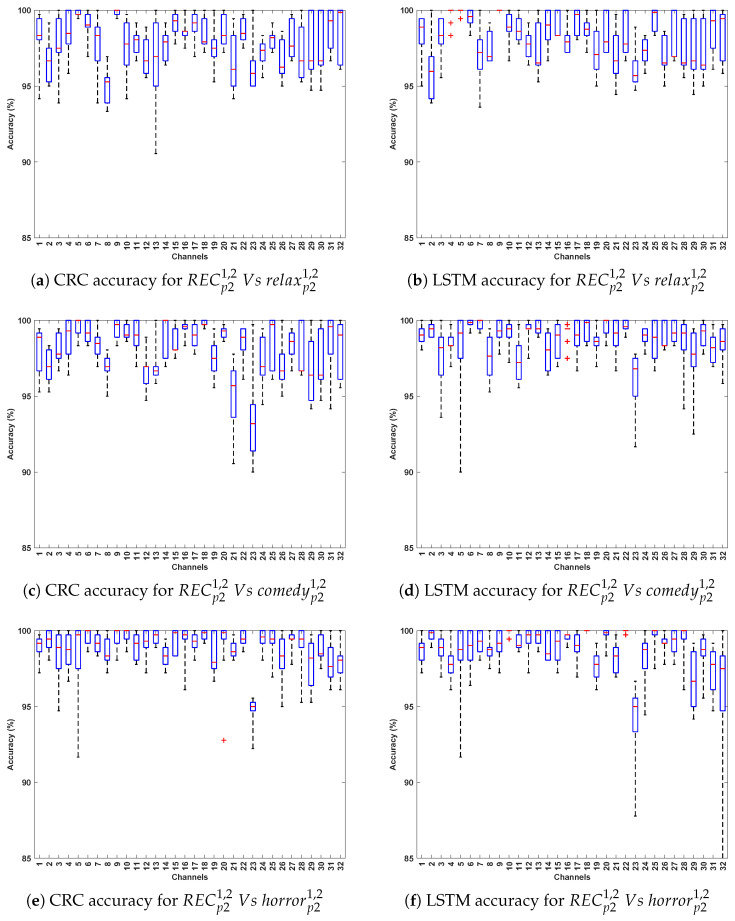
Illustrates the performance across 32 channels for PD detection for DCP 1 (Evaluation 2; presenting the two best-performing algorithms: CRC and LSTM for simplicity).

**Figure 9 bioengineering-12-01185-f009:**
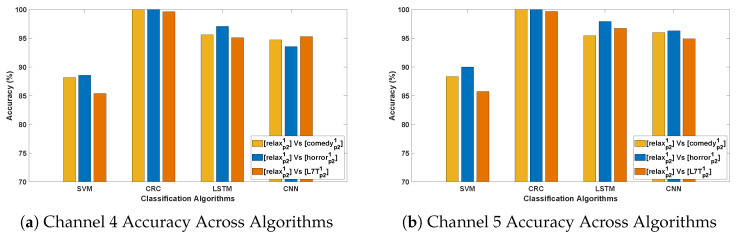
Illustrating the performance accuracy of the top four EEG channels across all four algorithms for DCP 2 (Evaluation 2; only the four best-performing channels are presented for simplicity).

**Figure 10 bioengineering-12-01185-f010:**
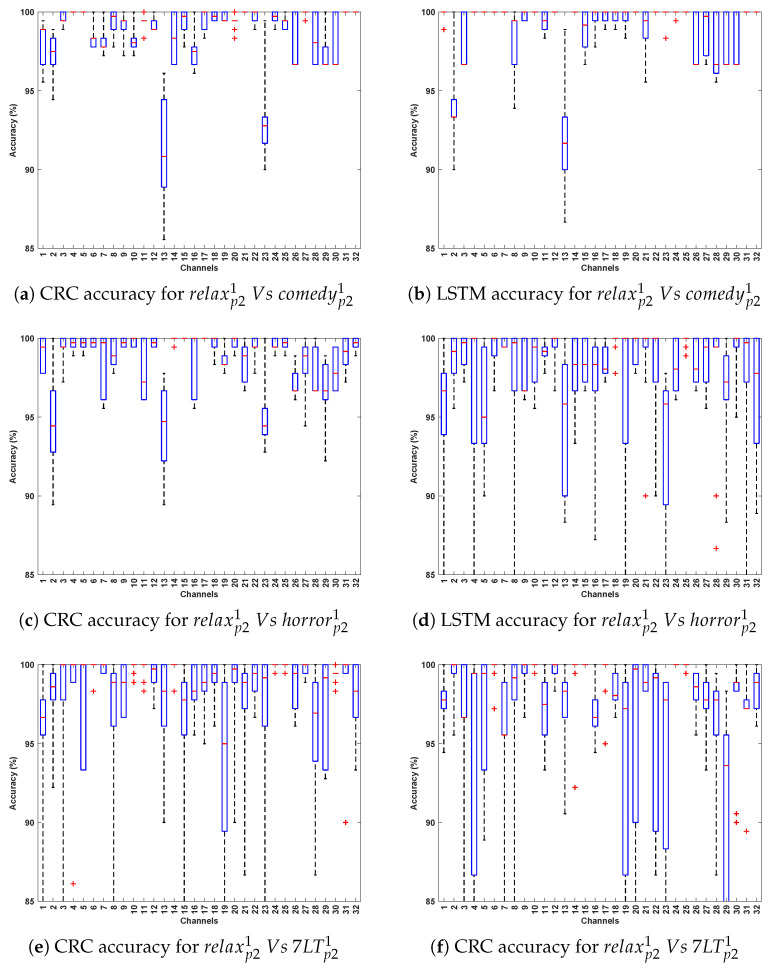
Illustrates the performance across 32 channels for PD detection for DCP 2 (Evaluation 2; presenting the two best-performing algorithms: CRC and LSTM for simplicity).

**Table 1 bioengineering-12-01185-t001:** Comprehensive overview of EEG recordings across two distinct Data Collection Protocols in the ParEEG database.

Data Acquisition Protocol	Stimuli		No. of Subject (30 HC & 30 PD)	Sample (Ch * Sample/Subject)	Total Sample (Sample * Subject)
	Description	Notation			
Data Collection Protocol 1	Resting State Eye Close	$rec_{p1}^{1}$	60	192 (32 * 06)	103,680 (1728 * 60)
		$rec_{p1}^{2}$	60	192 (32 * 06)	
	Relax State	$relax_{p1}^{1}$	60	192 (32 * 06)	
		$relax_{p1}^{2}$	60	192 (32 * 06)	
	Comedy State	$comedy_{p1}^{1}$	60	224 (32 * 07)	
		$comedy_{p1}^{2}$	60	224 (32 * 07)	
	Horror State	$horror_{p1}^{1}$	60	256 (32 * 08)	
		$horror_{p1}^{2}$	60	256 (32 * 08)	
Data Collection Protocol 2	Alpha Neumaric Flikering	$7LT_{p2}^{1}$	60	192 (32 * 06)	99,840 (1664 * 60)
		$7LT_{p2}^{2}$	60	192 (32 * 06)	
		$7LT_{p2}^{3}$	60	192 (32 * 06)	
		$7LT_{p2}^{4}$	60	192 (32 * 06)	
	Relax State	$relax_{p2}^{1}$	60	192 (32 * 06)	
	Comedy State	$comedy_{p2}^{1}$	60	256 (32 * 08)	
	Horror State	$horror_{p2}^{1}$	60	448 (32 * 14)	

**Table 2 bioengineering-12-01185-t002:** Average classification accuracy of four PD detection algorithms for within-stimulation Evaluation 1 using Data Collection Protocol 1.

Ch	${\mathrm{rec}}_{p1}^{1}$ Vs ${\mathrm{rec}}_{p1}^{2}$				${\mathrm{relax}}_{p1}^{1}$ Vs ${\mathrm{relax}}_{p1}^{2}$				${\mathrm{comedy}}_{p1}^{1}$ Vs ${\mathrm{comedy}}_{p1}^{2}$				${\mathrm{horror}}_{p1}^{1}$ Vs ${\mathrm{horror}}_{p1}^{2}$			
	SVM	CRC	LSTM	CNN	SVM	CRC	LSTM	CNN	SVM	CRC	LSTM	CNN	SVM	CRC	LSTM	CNN
1	79.0 ± 6.2	97.8 ± 2.0	99.3 ± 1.1	96.6 ± 1.4	78.1 ± 4.2	97.6 ± 1.7	98.8 ± 1.2	93.7 ± 2.4	78.6 ± 4.12	98.5 ± 1.8	98.2 ± 1.1	95.7 ± 2.7	78.9 ± 4.0	98.6 ± 1.5	98.3 ± 1.8	95.6 ± 2.9
2	69.1 ± 4.2	97.4 ± 1.5	98.9 ± 0.8	96.5 ± 1.1	68.3 ± 3.9	95.1 ± 2.8	98.8 ± 1.6	96.2 ± 1.8	68.7 ± 3.68	93.8 ± 3.1	99.7 ± 0.7	96.6 ± 2.4	70.1 ± 4.0	96.1 ± 2.7	100.0 ± 0.2	95.7 ± 1.9
3	78.9 ± 3.0	98.9 ± 1.4	97.8 ± 1.6	97.8 ± 1.0	80.3 ± 3.9	97.8 ± 2.0	97.8 ± 2.5	93.3 ± 3.0	80.3 ± 3.99	98.0 ± 1.5	95.7 ± 6.3	93.3 ± 7.7	79.7 ± 4.6	97.9 ± 1.3	94.8 ± 8.8	93.1 ± 7.3
4	83.1 ± 4.7	99.1 ± 1.3	99.7 ± 0.2	97.0 ± 1.3	84.4 ± 5.1	99.1 ± 1.1	98.3 ± 1.8	97.0 ± 2.1	85.1 ± 4.59	100.0 ± 0.2	97.0 ± 1.2	97.1 ± 2.1	85.7 ± 3.9	98.1 ± 1.9	96.1 ± 1.3	96.1 ± 1.9
5	83.8 ± 6.6	98.4 ± 1.4	97.4 ± 3.0	97.1 ± 3.2	90.8 ± 1.8	99.7 ± 0.3	97.4 ± 3.2	95.3 ± 3.7	91.0 ± 1.61	100.0 ± 0.0	97.6 ± 2.9	97.2 ± 2.8	90.3 ± 1.9	99.8 ± 0.5	98.3 ± 2.7	97.3 ± 1.7
6	79.5 ± 4.5	96.8 ± 2.5	99.6 ± 0.7	96.8 ± 1.8	82.0 ± 3.9	99.1 ± 0.4	99.5 ± 0.7	98.1 ± 1.3	82.8 ± 4.31	99.9 ± 0.2	98.7 ± 1.5	95.7 ± 3.2	83.2 ± 4.5	99.1 ± 1.3	98.3 ± 1.7	97.6 ± 2.1
7	76.6 ± 6.1	98.2 ± 1.7	99.1 ± 1.0	95.5 ± 1.5	78.1 ± 4.5	97.0 ± 2.4	99.6 ± 0.5	95.4 ± 1.9	78.5 ± 4.60	97.7 ± 1.3	98.8 ± 1.3	95.1 ± 1.6	78.8 ± 3.4	99.5 ± 0.9	99.0 ± 1.0	95.2 ± 2.7
8	77.6 ± 4.6	96.8 ± 1.9	97.7 ± 1.9	95.2 ± 2.6	77.7 ± 4.7	96.5 ± 1.9	97.8 ± 0.9	93.2 ± 2.1	78.2 ± 4.19	99.1 ± 0.7	99.0 ± 1.0	92.3 ± 2.5	79.0 ± 3.5	97.3 ± 2.5	99.2 ± 1.1	95.2 ± 3.4
9	67.6 ± 4.1	99.2 ± 0.7	98.7 ± 1.2	96.5 ± 1.2	74.3 ± 3.4	100.0 ± 0.1	99.7 ± 0.4	98.0 ± 1.3	74.7 ± 3.58	100.0 ± 0.0	99.8 ± 0.4	96.7 ± 1.6	74.7 ± 3.6	100.0 ± 0.0	99.2 ± 0.9	96.9 ± 2.9
10	56.9 ± 4.3	95.5 ± 2.9	98.5 ± 1.5	96.4 ± 2.1	59.0 ± 3.6	99.1 ± 1.3	99.6 ± 0.7	95.2 ± 1.7	59.0 ± 3.82	99.8 ± 0.3	99.8 ± 0.5	92.8 ± 2.9	58.9 ± 4.4	100.0 ± 0.0	99.8 ± 0.3	96.8 ± 2.2
11	68.2 ± 3.7	95.5 ± 1.8	99.3 ± 0.6	94.8 ± 2.0	67.8 ± 3.3	98.1 ± 1.3	97.6 ± 2.0	95.5 ± 1.5	68.3 ± 3.68	99.4 ± 1.0	98.7 ± 1.4	94.8 ± 2.6	68.2 ± 4.0	100.0 ± 0.2	99.5 ± 0.6	95.8 ± 2.2
12	75.2 ± 5.5	97.5 ± 2.1	99.0 ± 1.0	93.3 ± 2.3	76.5 ± 4.3	96.7 ± 1.8	99.4 ± 0.4	96.7 ± 2.0	76.8 ± 4.46	99.3 ± 0.4	100.0 ± 0.0	94.1 ± 2.6	77.1 ± 4.4	99.6 ± 0.9	99.4 ± 0.6	95.2 ± 1.7
13	69.7 ± 4.0	96.1 ± 4.3	98.3 ± 0.8	92.8 ± 4.3	69.3 ± 3.4	96.8 ± 2.3	99.3 ± 0.7	95.7 ± 2.1	69.4 ± 3.11	96.5 ± 1.8	99.3 ± 1.1	91.5 ± 1.6	68.7 ± 3.6	96.7 ± 1.2	99.5 ± 0.8	94.7 ± 1.6
14	78.0 ± 3.2	97.1 ± 1.9	98.0 ± 1.7	95.3 ± 1.3	79.0 ± 4.1	98.8 ± 1.6	99.0 ± 1.4	96.2 ± 1.4	79.6 ± 4.23	99.8 ± 1.8	99.2 ± 1.3	95.6 ± 1.5	79.1 ± 5.0	99.3 ± 0.8	98.7 ± 1.0	97.1 ± 1.7
15	74.7 ± 6.2	100.0 ± 0.0	99.2 ± 0.9	97.2 ± 1.9	77.5 ± 5.6	98.0 ± 1.7	97.0 ± 5.7	96.3 ± 1.2	77.6 ± 6.17	98.0 ± 1.7	99.0 ± 1.3	97.5 ± 1.1	76.6 ± 6.3	98.0 ± 1.7	96.7 ± 5.8	94.8 ± 1.9
16	73.2 ± 4.8	97.8 ± 1.6	99.6 ± 0.9	94.6 ± 2.1	72.2 ± 4.7	97.6 ± 1.0	97.3 ± 4.7	94.8 ± 2.3	73.3 ± 4.56	97.8 ± 2.6	97.9 ± 4.3	94.7 ± 1.7	74.3 ± 3.7	99.6 ± 1.0	98.3 ± 4.1	96.7 ± 2.2
17	72.3 ± 5.2	97.7 ± 1.9	97.4 ± 1.6	94.5 ± 2.2	73.3 ± 4.9	99.1 ± 1.0	98.1 ± 1.5	97.3 ± 1.5	73.1 ± 4.92	99.0 ± 1.6	98.3 ± 1.3	96.5 ± 1.7	73.1 ± 4.9	98.2 ± 1.9	97.6 ± 1.6	94.6 ± 1.9
18	70.5 ± 4.8	96.3 ± 1.9	100.0 ± 0.0	93.8 ± 2.8	68.5 ± 4.2	98.5 ± 0.9	98.8 ± 1.5	95.8 ± 2.2	69.7 ± 4.44	98.7 ± 2.2	99.7 ± 0.6	95.4 ± 2.0	69.3 ± 4.4	99.6 ± 0.4	100.0 ± 0.0	96.6 ± 1.9
19	72.0 ± 9.2	95.5 ± 3.6	99.1 ± 0.7	92.1 ± 2.3	73.3 ± 9.1	96.8 ± 2.6	94.2 ± 6.2	96.9 ± 1.8	74.6 ± 8.93	97.8 ± 1.9	98.0 ± 1.5	97.1 ± 1.6	74.1 ± 8.5	98.4 ± 1.2	97.0 ± 1.9	96.0 ± 2.0
20	81.6 ± 6.3	95.8 ± 2.9	99.4 ± 1.6	93.7 ± 1.1	84.7 ± 3.7	97.7 ± 1.8	98.1 ± 4.4	98.6 ± 1.4	85.3 ± 4.27	98.2 ± 2.4	99.5 ± 0.7	94.4 ± 2.6	85.0 ± 2.8	99.1 ± 1.9	98.8 ± 0.9	96.7 ± 2.2
21	68.0 ± 6.1	96.3 ± 2.9	99.5 ± 1.1	93.6 ± 2.6	75.2 ± 5.7	96.1 ± 2.6	98.6 ± 1.1	96.1 ± 1.5	76.2 ± 5.16	97.8 ± 1.9	99.3 ± 0.7	95.0 ± 2.4	77.5 ± 4.6	96.3 ± 2.4	97.0 ± 2.1	95.4 ± 1.8
22	78.6 ± 5.2	97.8 ± 1.5	99.9 ± 0.2	95.4 ± 2.0	81.1 ± 4.5	98.5 ± 1.6	99.6 ± 0.3	96.8 ± 1.3	82.9 ± 4.57	98.7 ± 1.2	99.5 ± 0.5	95.5 ± 2.4	83.0 ± 4.6	99.8 ± 0.3	98.7 ± 1.1	96.8 ± 1.5
23	58.7 ± 3.6	100.0 ± 0.1	99.4 ± 0.3	97.2 ± 1.4	64.1 ± 3.0	92.3 ± 3.1	91.8 ± 4.0	89.5 ± 3.7	65.0 ± 2.40	93.4 ± 2.8	94.8 ± 1.4	91.0 ± 3.7	66.1 ± 2.5	92.3 ± 2.9	92.6 ± 3.1	92.3 ± 3.3
24	69.8 ± 5.3	98.6 ± 1.3	99.7 ± 0.5	96.5 ± 1.1	77.4 ± 3.4	94.7 ± 2.3	97.0 ± 1.8	97.0 ± 2.7	77.7 ± 3.86	96.8 ± 2.1	94.9 ± 2.3	95.6 ± 1.8	77.7 ± 3.9	96.8 ± 1.2	98.5 ± 1.4	96.2 ± 1.8
25	80.6 ± 3.3	98.2 ± 1.6	99.2 ± 1.2	95.7 ± 1.2	79.6 ± 4.4	96.9 ± 1.3	97.8 ± 2.2	96.2 ± 1.6	78.7 ± 4.05	99.2 ± 1.3	98.8 ± 1.4	95.8 ± 2.2	78.8 ± 4.7	99.3 ± 0.7	99.1 ± 1.1	96.6 ± 1.7
26	72.4 ± 3.0	96.0 ± 2.8	99.2 ± 1.8	95.8 ± 1.7	69.0 ± 1.9	96.7 ± 1.2	99.5 ± 0.9	95.6 ± 2.0	68.9 ± 1.57	98.0 ± 1.7	99.2 ± 1.1	94.9 ± 2.5	69.3 ± 3.1	98.0 ± 1.7	98.7 ± 1.1	96.6 ± 2.0
27	77.4 ± 5.8	97.0 ± 2.2	100.0 ± 0.0	96.1 ± 2.2	83.9 ± 4.1	96.8 ± 2.7	97.7 ± 1.9	95.5 ± 1.3	85.1 ± 3.99	97.2 ± 1.9	98.1 ± 1.4	94.9 ± 2.0	86.0 ± 3.1	97.9 ± 2.4	98.2 ± 1.4	97.3 ± 1.8
28	79.2 ± 4.3	96.5 ± 2.7	99.4 ± 1.1	93.8 ± 2.6	77.0 ± 5.0	96.2 ± 2.0	97.0 ± 3.8	95.0 ± 1.9	79.0 ± 5.19	97.3 ± 1.9	99.0 ± 1.3	97.0 ± 1.4	80.5 ± 4.6	97.8 ± 1.6	99.4 ± 1.1	97.0 ± 1.5
29	68.2 ± 4.5	97.4 ± 1.8	98.8 ± 1.1	95.4 ± 1.7	72.3 ± 4.4	96.2 ± 2.1	95.2 ± 5.8	93.1 ± 2.8	73.0 ± 3.99	97.5 ± 2.0	96.5 ± 4.1	94.6 ± 1.4	73.1 ± 4.1	97.8 ± 1.6	98.4 ± 1.9	93.2 ± 2.1
30	70.5 ± 3.8	95.0 ± 3.9	98.7 ± 1.1	94.7 ± 1.4	74.1 ± 4.8	96.3 ± 2.0	98.1 ± 1.2	94.6 ± 2.2	74.5 ± 4.69	96.8 ± 2.3	97.7 ± 2.7	96.5 ± 1.9	75.9 ± 4.4	96.8 ± 1.6	95.8 ± 2.5	97.0 ± 2.6
31	75.2 ± 4.9	98.7 ± 1.7	98.0 ± 1.2	97.8 ± 1.9	77.2 ± 4.2	98.7 ± 1.7	97.6 ± 1.1	95.2 ± 2.1	78.3 ± 4.49	98.7 ± 1.7	98.1 ± 1.6	94.5 ± 2.8	78.6 ± 4.5	99.6 ± 0.7	95.5 ± 5.3	97.0 ± 1.1
32	81.0 ± 5.2	99.3 ± 1.1	99.1 ± 0.5	97.8 ± 1.0	81.3 ± 4.8	98.4 ± 1.7	99.1 ± 0.7	96.1 ± 2.4	81.5 ± 5.30	98.3 ± 1.7	98.1 ± 1.3	96.7 ± 1.8	81.0 ± 5.4	97.8 ± 2.3	97.9 ± 1.6	96.2 ± 1.9

**Table 3 bioengineering-12-01185-t003:** Average classification accuracy of four PD detection algorithms for within-stimulation Evaluation 1 using Data Collection Protocol 2.

Ch	$7{\mathrm{LT}}_{p2}^{1}$ Vs $7{\mathrm{LT}}_{p2}^{2}$				$7{\mathrm{LT}}_{p2}^{1}$ Vs $7{\mathrm{LT}}_{p2}^{3}$				$7{\mathrm{LT}}_{p2}^{1}$ Vs $7{\mathrm{LT}}_{p2}^{4}$			
	SVM	CRC	LSTM	CNN	SVM	CRC	LSTM	CNN	SVM	CRC	LSTM	CNN
1	82.0 ± 4.8	98.5 ± 1.7	96.5 ± 2.1	93.6 ± 2.3	80.3 ± 2.9	98.1 ± 1.9	97.6 ± 1.3	93.6 ± 2.5	83.5 ± 3.2	98.5 ± 1.7	96.8 ± 1.4	94.2 ± 2.5
2	72.0 ± 3.3	92.8 ± 3.0	98.5 ± 2.4	91.9 ± 4.1	70.3 ± 3.0	96.5 ± 2.6	99.5 ± 1.3	95.2 ± 1.8	72.3 ± 3.0	99.8 ± 0.4	98.4 ± 1.4	95.8 ± 2.1
3	80.2 ± 5.3	99.3 ± 0.7	99.8 ± 0.4	96.2 ± 2.3	77.7 ± 4.4	98.5 ± 1.1	99.2 ± 0.8	96.8 ± 1.3	77.8 ± 4.6	99.7 ± 0.5	99.0 ± 1.3	98.1 ± 0.6
4	88.5 ± 2.9	100.0 ± 0.0	99.3 ± 1.7	96.5 ± 1.8	88.0 ± 3.4	99.8 ± 0.5	98.7 ± 3.0	97.1 ± 1.7	88.7 ± 3.7	99.7 ± 0.7	97.2 ± 2.9	97.4 ± 2.3
5	88.7 ± 3.6	100.0 ± 0.0	98.3 ± 2.7	96.2 ± 3.0	87.5 ± 3.1	99.9 ± 0.4	96.9 ± 4.5	94.9 ± 3.7	88.2 ± 2.9	100.0 ± 0.0	96.3 ± 4.6	95.1 ± 3.9
6	81.4 ± 3.7	99.8 ± 0.3	98.6 ± 4.0	96.1 ± 1.8	79.5 ± 3.7	99.2 ± 0.5	99.1 ± 1.3	95.6 ± 3.0	80.2 ± 3.1	99.7 ± 0.3	97.7 ± 4.4	97.1 ± 1.5
7	84.9 ± 3.4	100.0 ± 0.2	99.1 ± 0.7	96.1 ± 2.2	82.0 ± 3.0	99.0 ± 0.8	99.2 ± 0.6	97.3 ± 1.4	83.1 ± 3.4	99.7 ± 0.4	99.8 ± 0.4	97.5 ± 1.3
8	83.3 ± 4.1	99.2 ± 1.3	97.0 ± 5.1	94.2 ± 5.9	82.3 ± 3.2	97.5 ± 1.0	97.5 ± 5.8	94.8 ± 6.3	83.6 ± 2.7	100.0 ± 0.0	97.3 ± 5.5	93.6 ± 5.9
9	75.2 ± 3.8	99.6 ± 0.6	98.4 ± 1.6	96.5 ± 2.4	73.9 ± 3.2	99.6 ± 0.6	98.5 ± 1.2	96.2 ± 2.3	74.0 ± 3.3	99.5 ± 0.6	99.1 ± 1.5	95.9 ± 2.4
10	60.7 ± 3.4	100.0 ± 0.0	100.0 ± 0.0	96.3 ± 1.3	59.7 ± 4.3	99.3 ± 0.5	99.7 ± 0.5	96.7 ± 1.4	59.6 ± 4.5	99.8 ± 0.4	99.3 ± 0.9	95.8 ± 1.9
11	65.0 ± 2.8	99.9 ± 0.2	99.6 ± 0.4	92.7 ± 2.7	65.1 ± 2.8	99.3 ± 0.5	99.4 ± 0.8	93.5 ± 3.0	65.7 ± 2.3	99.9 ± 0.2	98.3 ± 1.5	93.3 ± 2.0
12	75.3 ± 7.0	99.5 ± 0.7	98.6 ± 1.1	94.2 ± 4.1	72.4 ± 4.9	99.2 ± 0.5	98.8 ± 0.6	96.4 ± 1.7	74.1 ± 4.9	99.6 ± 0.3	99.5 ± 0.7	94.0 ± 1.8
13	69.6 ± 2.8	92.2 ± 3.6	96.8 ± 4.8	89.2 ± 5.5	67.8 ± 2.9	90.6 ± 3.5	95.1 ± 4.9	91.7 ± 4.2	69.1 ± 3.7	92.2 ± 4.8	93.8 ± 4.0	93.5 ± 3.3
14	80.0 ± 4.4	100.0 ± 0.0	100.0 ± 0.2	97.2 ± 1.1	80.0 ± 4.4	98.8 ± 1.0	98.5 ± 1.5	95.3 ± 1.4	80.0 ± 4.4	99.1 ± 1.0	98.3 ± 1.5	95.8 ± 2.7
15	78.0 ± 5.9	100.0 ± 0.0	97.6 ± 4.9	96.3 ± 2.2	78.2 ± 6.2	99.0 ± 1.1	97.4 ± 3.8	96.2 ± 2.0	80.3 ± 6.6	100.0 ± 0.2	100.0 ± 0.0	96.7 ± 2.2
16	73.6 ± 5.0	99.6 ± 0.6	98.8 ± 0.8	93.8 ± 2.9	71.3 ± 2.8	98.6 ± 1.0	96.4 ± 4.6	95.2 ± 2.1	71.5 ± 3.0	97.7 ± 1.5	97.8 ± 2.0	95.0 ± 2.7
17	76.0 ± 4.6	100.0 ± 0.0	100.0 ± 0.0	97.5 ± 2.1	76.3 ± 5.1	99.8 ± 0.4	97.1 ± 4.7	97.8 ± 1.2	77.0 ± 4.9	100.0 ± 0.0	99.4 ± 0.7	94.1 ± 3.0
18	70.1 ± 5.1	100.0 ± 0.0	100.0 ± 0.2	95.5 ± 2.6	67.9 ± 4.3	99.9 ± 0.2	99.0 ± 1.3	97.3 ± 1.6	68.2 ± 4.6	99.8 ± 0.4	99.5 ± 0.7	95.6 ± 2.5
19	67.8 ± 8.7	99.6 ± 0.2	96.7 ± 4.3	94.6 ± 1.8	66.8 ± 6.8	99.8 ± 0.5	98.5 ± 1.5	97.6 ± 1.6	67.2 ± 7.4	99.8 ± 0.4	97.8 ± 4.3	96.4 ± 1.8
20	83.0 ± 5.6	100.0 ± 0.0	99.8 ± 0.5	96.9 ± 1.7	80.8 ± 3.4	99.6 ± 0.4	98.9 ± 3.0	98.1 ± 1.2	81.8 ± 3.9	99.7 ± 0.4	97.5 ± 4.6	95.8 ± 2.6
21	85.3 ± 5.1	98.6 ± 1.8	96.3 ± 4.0	91.6 ± 4.9	85.6 ± 5.0	99.9 ± 0.4	99.0 ± 3.0	95.8 ± 3.7	86.3 ± 7.0	99.9 ± 0.4	99.0 ± 3.0	95.1 ± 3.9
22	83.5 ± 7.3	100.0 ± 0.2	99.9 ± 0.2	96.6 ± 1.6	81.5 ± 4.6	99.3 ± 1.1	98.7 ± 1.1	96.2 ± 1.5	83.1 ± 3.8	99.2 ± 0.7	99.3 ± 0.7	95.3 ± 2.8
23	67.2 ± 5.1	98.8 ± 2.5	97.5 ± 2.8	95.2 ± 2.1	71.1 ± 4.6	93.8 ± 3.3	94.5 ± 3.2	96.1 ± 2.1	73.0 ± 4.2	93.7 ± 2.9	95.5 ± 3.2	96.3 ± 3.3
24	81.3 ± 4.2	100.0 ± 0.0	100.0 ± 0.0	96.0 ± 2.1	81.1 ± 3.3	99.9 ± 0.4	99.6 ± 0.5	96.8 ± 1.6	81.2 ± 4.3	99.7 ± 0.6	97.9 ± 1.7	96.6 ± 2.2
25	84.2 ± 3.3	99.7 ± 0.4	99.2 ± 0.8	95.9 ± 2.7	82.1 ± 4.4	99.6 ± 0.5	99.6 ± 0.3	97.5 ± 1.3	86.0 ± 3.0	99.8 ± 0.4	99.7 ± 0.5	95.3 ± 2.8
26	67.5 ± 4.7	97.8 ± 1.5	99.2 ± 0.9	95.8 ± 2.4	68.1 ± 5.9	97.5 ± 1.5	98.7 ± 1.0	96.9 ± 2.3	71.5 ± 4.8	96.5 ± 1.3	96.5 ± 1.8	93.3 ± 3.1
27	86.0 ± 2.1	98.8 ± 1.5	99.7 ± 0.5	96.0 ± 2.1	85.3 ± 2.3	99.9 ± 0.4	99.1 ± 0.8	96.0 ± 1.8	85.3 ± 2.3	100.0 ± 0.2	99.1 ± 1.4	96.0 ± 1.2
28	82.0 ± 3.8	97.8 ± 1.9	96.1 ± 3.0	95.7 ± 3.4	83.6 ± 4.4	98.0 ± 1.7	98.5 ± 2.9	95.8 ± 3.6	86.3 ± 4.2	97.4 ± 1.3	97.0 ± 2.5	96.2 ± 2.6
29	76.0 ± 4.7	98.0 ± 1.6	97.2 ± 2.6	95.3 ± 1.7	77.4 ± 6.2	98.0 ± 1.7	98.1 ± 1.6	95.6 ± 1.8	78.9 ± 5.9	97.7 ± 1.5	96.4 ± 2.7	95.5 ± 1.6
30	78.7 ± 4.8	98.2 ± 1.6	96.8 ± 5.1	97.4 ± 1.4	77.5 ± 3.7	97.7 ± 1.4	99.5 ± 0.8	97.7 ± 1.2	79.2 ± 3.6	97.6 ± 1.3	98.7 ± 1.3	96.3 ± 2.2
31	82.7 ± 3.6	100.0 ± 0.2	97.6 ± 2.6	96.5 ± 2.0	84.5 ± 4.3	99.8 ± 0.3	98.0 ± 2.8	96.1 ± 2.2	85.1 ± 4.1	100.0 ± 0.2	97.7 ± 2.7	96.7 ± 1.5
32	83.2 ± 3.9	100.0 ± 0.0	99.7 ± 0.5	97.0 ± 2.0	84.1 ± 4.0	99.7 ± 0.3	99.6 ± 0.5	95.5 ± 1.8	85.3 ± 3.3	99.6 ± 0.5	98.7 ± 1.1	96.0 ± 2.3

**Table 4 bioengineering-12-01185-t004:** Average Classification Accuracy of Four PD Detection Algorithms for Cross-Stimulation Evaluation 2 Using Data Collection Protocol 1.

Ch	${\mathrm{rec}}_{p1}^{1,2}$ Vs ${\mathrm{relax}}_{p1}^{1,2}$				${\mathrm{rec}}_{p1}^{1,2}$ Vs ${\mathrm{comedy}}_{p1}^{1,2}$				${\mathrm{rec}}_{p1}^{1,2}$ Vs ${\mathrm{horror}}_{p1}^{1,2}$			
	SVM	CRC	LSTM	CNN	SVM	CRC	LSTM	CNN	SVM	CRC	LSTM	CNN
1	79.2 ± 5.1	98.1 ± 1.9	99.0 ± 0.6	97.5 ± 1.4	79.8 ± 5.0	98.1 ± 1.7	96.9 ± 6.6	97.4 ± 0.9	79.6 ± 4.8	97.9 ± 1.7	97.1 ± 5.0	97.6 ± 1.6
2	69.0 ± 4.0	96.6 ± 1.4	99.4 ± 0.4	97.8 ± 0.8	69.0 ± 4.0	96.1 ± 2.0	99.3 ± 0.6	97.8 ± 0.9	69.6 ± 3.7	97.0 ± 1.1	99.7 ± 0.4	98.4 ± 0.9
3	80.3 ± 3.1	97.6 ± 1.9	97.6 ± 1.8	96.3 ± 1.6	80.3 ± 3.1	98.2 ± 1.2	98.2 ± 1.7	97.6 ± 1.4	80.8 ± 3.4	98.1 ± 1.0	98.8 ± 0.9	97.8 ± 1.3
4	84.0 ± 4.9	98.5 ± 1.4	98.5 ± 0.7	98.5 ± 1.2	84.6 ± 4.5	99.8 ± 0.6	98.7 ± 1.0	98.7 ± 0.6	85.0 ± 4.2	98.7 ± 1.4	97.9 ± 1.2	98.3 ± 1.1
5	88.6 ± 3.8	99.8 ± 0.2	98.1 ± 2.9	97.8 ± 2.7	89.7 ± 2.9	100.0 ± 0.2	98.4 ± 2.5	98.5 ± 2.3	89.9 ± 3.0	99.6 ± 0.6	98.3 ± 2.4	98.3 ± 2.5
6	83.0 ± 3.1	99.0 ± 0.9	99.8 ± 0.3	98.4 ± 0.4	83.5 ± 3.2	99.5 ± 0.6	99.6 ± 0.5	98.6 ± 0.9	83.5 ± 3.6	99.3 ± 0.7	98.9 ± 1.2	99.1 ± 0.7
7	78.3 ± 4.3	97.9 ± 1.8	99.7 ± 0.4	98.2 ± 1.4	79.1 ± 4.1	97.1 ± 1.9	99.1 ± 0.6	97.8 ± 1.2	79.8 ± 3.7	98.4 ± 0.9	99.2 ± 0.7	98.4 ± 1.0
8	77.6 ± 4.7	95.0 ± 1.1	97.6 ± 1.3	95.0 ± 0.9	77.9 ± 4.5	97.5 ± 1.0	98.6 ± 0.9	96.5 ± 0.7	78.3 ± 4.1	96.9 ± 0.9	98.7 ± 0.6	97.0 ± 2.0
9	72.0 ± 3.0	99.8 ± 0.2	99.3 ± 0.7	98.6 ± 0.8	72.4 ± 3.0	100.0 ± 0.0	99.6 ± 0.6	98.8 ± 0.7	73.3 ± 3.5	99.5 ± 0.7	99.1 ± 0.8	98.6 ± 0.8
10	58.4 ± 3.7	97.6 ± 1.8	99.3 ± 0.8	97.8 ± 1.0	58.7 ± 4.0	98.8 ± 1.2	99.7 ± 0.5	98.2 ± 1.4	58.8 ± 4.1	99.2 ± 0.5	100.0 ± 0.2	98.4 ± 1.1
11	68.2 ± 3.3	97.8 ± 0.8	97.5 ± 1.4	96.8 ± 1.4	68.3 ± 3.8	98.8 ± 0.8	99.0 ± 0.8	96.2 ± 2.0	68.5 ± 3.9	99.0 ± 1.0	99.2 ± 0.5	96.7 ± 1.7
12	76.2 ± 4.8	97.0 ± 1.2	99.4 ± 0.7	97.1 ± 2.0	76.5 ± 5.0	97.7 ± 0.9	99.1 ± 0.9	96.3 ± 2.4	77.0 ± 5.0	96.8 ± 1.2	99.4 ± 0.8	96.8 ± 1.5
13	70.2 ± 3.6	96.7 ± 3.0	99.6 ± 0.4	96.5 ± 2.4	69.9 ± 3.5	97.6 ± 1.8	99.6 ± 0.4	95.4 ± 1.6	69.6 ± 3.5	97.2 ± 1.5	99.5 ± 0.5	96.2 ± 1.5
14	79.0 ± 3.9	97.7 ± 1.1	98.0 ± 1.1	97.9 ± 1.3	79.5 ± 3.8	98.9 ± 1.1	98.3 ± 0.7	97.9 ± 1.6	79.2 ± 4.1	99.0 ± 1.4	99.0 ± 0.9	98.4 ± 1.0
15	76.9 ± 6.1	99.1 ± 0.8	98.7 ± 1.1	98.4 ± 0.6	77.3 ± 5.6	99.0 ± 0.8	99.4 ± 0.7	98.4 ± 0.8	77.4 ± 5.8	98.6 ± 1.0	99.0 ± 1.0	98.2 ± 1.0
16	74.1 ± 5.3	98.6 ± 0.6	99.2 ± 0.6	96.1 ± 2.3	74.3 ± 5.4	98.0 ± 0.9	99.4 ± 1.1	96.0 ± 2.4	75.4 ± 5.6	99.5 ± 0.5	99.5 ± 0.3	96.4 ± 1.4
17	73.0 ± 4.9	99.0 ± 1.0	98.9 ± 1.0	98.7 ± 0.7	73.2 ± 4.9	99.3 ± 0.8	99.3 ± 0.5	96.9 ± 1.5	73.5 ± 4.7	99.0 ± 0.8	98.8 ± 1.0	98.1 ± 1.0
18	69.9 ± 4.6	98.3 ± 1.0	99.5 ± 0.7	98.1 ± 1.1	70.2 ± 4.5	98.8 ± 0.9	99.7 ± 0.3	97.3 ± 1.0	70.6 ± 4.9	99.8 ± 0.2	100.0 ± 0.0	96.4 ± 1.6
19	73.1 ± 9.9	97.6 ± 1.3	98.6 ± 0.9	97.4 ± 1.1	73.5 ± 10.0	97.4 ± 1.8	98.4 ± 1.2	96.6 ± 1.4	74.0 ± 9.6	97.5 ± 1.1	97.8 ± 0.9	97.2 ± 1.5
20	85.6 ± 3.8	98.6 ± 1.1	99.6 ± 0.6	96.4 ± 1.5	85.8 ± 4.4	98.5 ± 1.4	99.0 ± 2.1	97.3 ± 1.5	86.0 ± 4.0	99.2 ± 0.4	99.7 ± 0.5	96.3 ± 1.1
21	72.5 ± 6.7	96.6 ± 1.8	99.0 ± 1.0	97.1 ± 1.5	73.5 ± 6.4	96.9 ± 1.6	98.8 ± 0.5	97.4 ± 1.8	74.2 ± 6.3	95.0 ± 2.4	98.2 ± 0.9	96.9 ± 1.5
22	82.3 ± 3.7	98.6 ± 0.8	99.6 ± 0.4	97.1 ± 1.9	82.5 ± 4.5	98.2 ± 1.3	99.5 ± 0.4	97.5 ± 1.8	83.1 ± 4.6	98.5 ± 1.1	100.0 ± 0.1	97.5 ± 1.3
23	63.0 ± 3.7	96.2 ± 1.6	96.1 ± 1.8	95.1 ± 1.6	63.5 ± 3.0	96.0 ± 1.2	94.7 ± 0.9	94.8 ± 1.8	64.3 ± 2.6	93.5 ± 2.9	94.1 ± 2.4	95.8 ± 2.0
24	75.6 ± 4.1	97.1 ± 0.9	98.9 ± 0.5	98.2 ± 0.8	76.5 ± 4.4	97.3 ± 0.8	99.5 ± 0.6	98.1 ± 1.1	76.8 ± 4.1	97.2 ± 1.5	98.3 ± 1.6	98.2 ± 0.7
25	81.0 ± 3.6	98.1 ± 0.6	98.6 ± 1.4	96.3 ± 1.3	81.0 ± 3.5	99.5 ± 0.7	99.2 ± 1.0	97.0 ± 2.0	81.3 ± 4.0	98.8 ± 1.7	99.6 ± 0.8	96.3 ± 2.4
26	70.8 ± 2.0	96.8 ± 1.3	99.0 ± 0.8	97.0 ± 1.3	71.0 ± 2.1	97.0 ± 1.2	98.3 ± 1.5	97.6 ± 1.0	70.0 ± 2.0	97.0 ± 1.4	99.1 ± 0.6	97.3 ± 1.1
27	82.9 ± 3.8	98.0 ± 1.3	99.3 ± 0.7	97.1 ± 0.7	83.8 ± 3.8	98.1 ± 1.6	99.4 ± 0.6	96.4 ± 1.5	84.2 ± 3.3	98.4 ± 1.0	99.2 ± 0.8	97.6 ± 0.9
28	78.6 ± 4.9	97.1 ± 1.8	98.6 ± 1.7	97.1 ± 1.6	79.2 ± 4.4	97.5 ± 1.7	99.1 ± 1.4	97.5 ± 1.4	78.7 ± 4.3	97.9 ± 1.6	99.5 ± 1.1	97.3 ± 1.5
29	71.5 ± 4.0	97.4 ± 2.1	97.5 ± 2.0	95.9 ± 0.8	72.4 ± 4.1	97.4 ± 2.0	97.7 ± 1.5	96.4 ± 1.4	72.8 ± 3.9	97.0 ± 2.2	96.9 ± 1.7	96.4 ± 1.2
30	73.3 ± 4.3	97.4 ± 2.0	99.1 ± 0.7	97.1 ± 1.3	73.8 ± 4.5	97.5 ± 2.0	98.9 ± 0.9	98.3 ± 1.2	74.4 ± 4.4	97.2 ± 2.0	98.4 ± 1.4	97.2 ± 1.6
31	77.0 ± 4.3	98.8 ± 1.3	98.2 ± 0.9	97.6 ± 1.3	77.3 ± 4.0	98.6 ± 1.6	98.0 ± 1.2	98.4 ± 1.4	77.7 ± 4.1	98.5 ± 2.1	97.4 ± 1.4	98.6 ± 0.7
32	81.0 ± 5.3	98.5 ± 1.8	97.0 ± 4.9	98.6 ± 0.7	81.5 ± 5.2	98.4 ± 1.8	98.1 ± 1.1	98.2 ± 1.4	81.1 ± 5.4	98.1 ± 1.9	95.7 ± 5.6	97.6 ± 1.0

**Table 5 bioengineering-12-01185-t005:** Average classification accuracy of four PD detection algorithms for cross-stimulation Evaluation 2 using Data Collection Protocol 2.

Ch	${\mathrm{relax}}_{p1}^{1}$ Vs ${\mathrm{comedy}}_{p2}^{1}$				${\mathrm{relax}}_{p1}^{1}$ Vs ${\mathrm{chorror}}_{p2}^{1}$				${\mathrm{relax}}_{p1}^{1}$ Vs $7{\mathrm{LT}}_{p2}^{1}$			
	SVM	CRC	LSTM	CNN	SVM	CRC	LSTM	CNN	SVM	CRC	LSTM	CNN
1	82.8 ± 2.7	97.8 ± 1.5	95.3 ± 4.4	94.9 ± 1.7	82.4 ± 4.7	99.8 ± 0.5	95.9 ± 4.1	96.3 ± 1.6	79.5 ± 4.2	98.9 ± 1.0	97.6 ± 1.5	94.9 ± 2.0
2	70.8 ± 3.4	97.3 ± 1.4	98.7 ± 1.5	95.7 ± 1.8	72.0 ± 3.2	93.8 ± 2.9	98.2 ± 2.2	90.0 ± 3.1	70.4 ± 3.7	94.6 ± 3.1	99.4 ± 1.3	95.1 ± 3.1
3	78.4 ± 4.3	99.6 ± 0.4	96.4 ± 8.7	94.7 ± 7.6	81.1 ± 5.2	98.0 ± 1.7	97.2 ± 6.5	95.4 ± 6.2	77.7 ± 4.8	99.2 ± 1.0	95.8 ± 6.7	94.6 ± 7.0
4	88.2 ± 3.2	100.0 ± 0.0	95.6 ± 6.2	94.7 ± 5.7	88.6 ± 2.5	100.0 ± 0.0	97.1 ± 5.5	93.6 ± 6.3	85.4 ± 4.3	99.6 ± 0.5	95.1 ± 6.9	95.3 ± 5.1
5	88.3 ± 2.8	100.0 ± 0.0	95.4 ± 3.7	96.0 ± 4.0	90.0 ± 2.7	100.0 ± 0.0	97.9 ± 3.0	96.3 ± 2.8	85.7 ± 2.9	99.7 ± 0.4	96.7 ± 4.2	94.9 ± 4.5
6	79.3 ± 3.8	98.4 ± 0.9	99.3 ± 1.1	96.5 ± 1.2	81.3 ± 3.6	100.0 ± 0.0	99.8 ± 0.5	97.2 ± 1.5	80.1 ± 4.0	99.7 ± 0.3	99.7 ± 0.8	97.3 ± 1.0
7	84.0 ± 2.9	98.2 ± 0.9	95.7 ± 8.1	94.9 ± 7.0	86.0 ± 3.4	100.0 ± 0.0	96.6 ± 6.6	95.3 ± 4.5	82.7 ± 2.7	98.6 ± 1.9	93.3 ± 8.0	94.3 ± 6.0
8	83.3 ± 3.1	99.4 ± 0.8	97.2 ± 5.9	95.0 ± 5.9	84.2 ± 3.6	98.2 ± 2.3	96.4 ± 5.7	93.6 ± 6.2	78.4 ± 2.8	98.9 ± 0.9	96.8 ± 6.8	93.9 ± 6.7
9	74.0 ± 3.4	99.2 ± 0.8	97.8 ± 1.6	96.1 ± 1.5	75.3 ± 3.9	99.8 ± 0.3	98.4 ± 1.6	97.3 ± 2.2	74.0 ± 3.4	99.7 ± 0.3	99.5 ± 1.0	97.2 ± 1.4
10	60.7 ± 3.1	98.3 ± 0.9	98.7 ± 1.6	96.8 ± 1.5	61.3 ± 2.8	100.0 ± 0.0	99.8 ± 0.4	95.8 ± 2.1	61.1 ± 3.3	99.8 ± 0.3	99.9 ± 0.2	96.9 ± 1.3
11	66.1 ± 4.8	99.3 ± 0.5	99.1 ± 0.7	95.8 ± 2.0	66.0 ± 4.7	99.4 ± 0.7	99.7 ± 0.6	96.6 ± 1.4	65.4 ± 3.3	97.7 ± 1.7	97.2 ± 2.0	92.8 ± 1.4
12	74.3 ± 4.5	99.2 ± 0.5	99.3 ± 1.2	98.1 ± 1.7	77.6 ± 5.8	100.0 ± 0.0	99.3 ± 0.9	96.1 ± 1.8	73.7 ± 2.6	99.7 ± 0.3	99.7 ± 0.5	96.8 ± 1.5
13	69.2 ± 4.1	91.4 ± 3.5	94.7 ± 4.1	92.3 ± 2.9	70.7 ± 3.1	92.1 ± 3.5	97.3 ± 3.2	87.5 ± 3.0	69.6 ± 3.2	94.3 ± 3.1	97.4 ± 2.6	94.4 ± 1.9
14	80.0 ± 4.4	98.3 ± 1.8	98.0 ± 2.2	94.8 ± 2.9	80.0 ± 4.4	100.0 ± 0.0	99.8 ± 0.5	96.5 ± 1.6	79.7 ± 4.0	99.9 ± 0.2	99.2 ± 2.3	97.8 ± 1.7
15	79.6 ± 6.0	99.3 ± 0.8	98.5 ± 1.4	97.0 ± 2.5	78.3 ± 5.9	98.9 ± 1.1	95.3 ± 6.0	95.8 ± 2.5	76.4 ± 6.1	100.0 ± 0.0	96.7 ± 6.7	97.4 ± 0.7
16	72.0 ± 3.2	97.5 ± 1.2	97.1 ± 3.7	96.9 ± 1.9	74.4 ± 4.7	99.6 ± 0.7	98.5 ± 1.5	96.4 ± 2.5	71.1 ± 3.4	98.7 ± 1.9	97.1 ± 1.7	95.1 ± 1.0
17	76.9 ± 5.3	99.6 ± 0.7	98.3 ± 0.9	97.0 ± 1.7	76.2 ± 5.1	99.7 ± 0.5	98.7 ± 1.5	96.8 ± 1.7	74.7 ± 5.7	100.0 ± 0.0	99.3 ± 1.5	96.4 ± 1.8
18	70.1 ± 5.1	99.7 ± 0.3	99.7 ± 0.7	96.8 ± 2.4	71.8 ± 5.0	99.7 ± 0.5	98.9 ± 1.4	95.4 ± 2.3	69.4 ± 5.0	99.7 ± 0.5	98.4 ± 1.1	96.1 ± 1.5
19	68.7 ± 7.8	99.7 ± 0.3	96.1 ± 6.3	97.4 ± 2.0	69.1 ± 9.2	99.7 ± 0.6	93.9 ± 5.4	96.9 ± 1.5	68.7 ± 8.2	98.6 ± 0.7	93.7 ± 7.0	96.5 ± 2.0
20	82.0 ± 2.7	99.3 ± 0.4	99.3 ± 0.9	97.1 ± 1.2	84.7 ± 3.3	100.0 ± 0.0	97.8 ± 3.8	97.2 ± 1.9	82.1 ± 3.1	99.8 ± 0.4	95.8 ± 6.2	96.4 ± 1.7
21	86.2 ± 5.7	100.0 ± 0.0	98.6 ± 3.0	94.2 ± 4.3	87.0 ± 3.7	98.9 ± 1.5	97.6 ± 3.8	94.4 ± 3.8	82.5 ± 2.7	98.6 ± 1.2	97.2 ± 5.8	93.9 ± 6.0
22	85.1 ± 2.9	99.8 ± 0.4	98.0 ± 3.5	95.1 ± 2.4	88.3 ± 3.6	100.0 ± 0.0	99.0 ± 1.1	97.3 ± 1.6	83.3 ± 4.2	99.4 ± 0.8	95.7 ± 5.1	96.8 ± 2.7
23	74.2 ± 5.3	93.2 ± 2.9	93.2 ± 5.5	92.8 ± 5.3	71.3 ± 8.1	99.8 ± 0.5	95.9 ± 6.4	93.9 ± 5.6	70.0 ± 7.4	95.1 ± 2.1	94.4 ± 6.4	93.7 ± 5.4
24	82.8 ± 4.4	99.7 ± 0.4	98.3 ± 1.5	97.1 ± 1.7	81.9 ± 4.5	99.9 ± 0.2	99.9 ± 0.2	95.6 ± 1.2	80.2 ± 4.9	99.6 ± 0.4	100.0 ± 0.0	97.1 ± 1.9
25	84.6 ± 5.7	99.3 ± 0.4	99.8 ± 0.4	96.6 ± 1.8	86.1 ± 5.0	100.0 ± 0.0	99.9 ± 0.2	96.4 ± 1.7	82.7 ± 5.4	99.6 ± 0.5	99.9 ± 0.2	96.4 ± 1.7
26	72.1 ± 7.3	97.9 ± 1.7	98.4 ± 1.3	96.7 ± 2.0	71.1 ± 7.7	98.0 ± 1.7	98.7 ± 1.5	96.3 ± 1.9	69.8 ± 4.6	97.1 ± 0.9	98.3 ± 1.2	95.9 ± 1.2
27	85.3 ± 2.3	99.9 ± 0.2	98.7 ± 1.6	96.3 ± 1.3	86.8 ± 1.8	98.8 ± 1.5	99.7 ± 0.5	95.4 ± 2.1	83.3 ± 2.2	98.4 ± 1.6	97.6 ± 2.0	96.1 ± 2.5
28	85.8 ± 4.4	98.3 ± 1.7	97.4 ± 4.6	95.6 ± 5.2	82.7 ± 3.3	97.8 ± 1.9	95.6 ± 4.2	93.8 ± 6.1	80.7 ± 3.4	97.8 ± 1.5	95.9 ± 4.2	95.3 ± 4.1
29	78.8 ± 5.8	97.6 ± 1.4	96.1 ± 4.0	95.9 ± 1.9	76.9 ± 4.8	98.0 ± 1.7	97.4 ± 3.0	95.8 ± 1.7	75.5 ± 3.6	96.4 ± 2.1	90.4 ± 6.5	94.4 ± 1.7
30	78.1 ± 3.2	97.9 ± 1.7	99.3 ± 1.5	98.1 ± 1.6	79.1 ± 4.9	97.9 ± 1.7	99.4 ± 0.5	97.6 ± 1.8	75.8 ± 3.4	97.9 ± 1.3	97.2 ± 3.5	97.3 ± 1.5
31	85.7 ± 2.3	100.0 ± 0.0	96.4 ± 6.3	96.6 ± 2.0	83.9 ± 2.9	100.0 ± 0.0	97.1 ± 5.9	97.2 ± 1.9	80.9 ± 2.8	99.0 ± 1.0	95.1 ± 6.5	95.9 ± 3.2
32	85.5 ± 2.7	100.0 ± 0.0	96.3 ± 4.1	98.2 ± 1.1	85.4 ± 3.3	100.0 ± 0.0	97.9 ± 2.0	97.4 ± 1.8	80.9 ± 3.7	99.6 ± 0.5	98.4 ± 1.4	97.1 ± 1.2

## Data Availability

The original contributions presented in this study are included in the article. Further inquiries can be directed to the corresponding author.

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
