# Peer review of "Stimulus-Evoked Brain Signals for Parkinson’s Detection: A Comprehensive Benchmark Performance Analysis on Cross-Stimulation and Channel-Wise Experiments"

_bioengineering, 2025, doi:10.3390/bioengineering12111185_

Round 1

Reviewer 1 Report

Comments and Suggestions for Authors

Summary

This article introduces a novel cross-stimulation evaluation framework to assess the generalizability of EEG-based Parkinson's Disease (PD) detection algorithms, moving beyond traditional resting-state analysis. The authors present the new ParEEG database, comprising over 200,000 EEG samples from 60 subjects (30 healthy, 30 PD) recorded under two types of visual stimuli: Resting-State Visual Evoked Potentials (RSVEP) and Steady-State Visually Evoked Potentials (SSVEP), which include emotionally engaging videos (relaxation, comedy, horror). The study performs a detailed channel-wise analysis using two handcrafted (SVM, CRC) and two deep learning (LSTM, CNN) classifiers. The key finding is that CRC and LSTM models consistently achieved high accuracy (often 95-100%), and that specific channels in the frontal, fronto-central, and central-parietal regions (e.g., F7, C3, Cp2) were the most discriminative for PD detection across all stimulus conditions. The study is generally interesting but also has some drawbacks.

Comment:

  1. The RSVEP and SSVEP dataset are good for mechanism study, but not optimal for disease detection, especially movement disorders like PD.
  2. The quantitative results should be added to the abstract, as well as the significance level.
  3. The introduction should be more concise and focus.
  4. Line 85, the authors say “Unlike conventional approaches that utilize all EEG channels collectively, this method enables the identification of the most discriminative individual channels for PD detection”. A lot methods analyzed the signal from channels separately, what do the authors mean by the word “collecetively”?
  5. one-minute baseline EEG recording may not enough for resting state activity monitoring, also, the sampling frequency is close to the lower limit required for the analysis.
  6. Do the authors using ICA analysis to remove the artefacts? How do the authors deal with the eye movement and head motion? The EEG artefact shoud be a problem in movement disorders like PD.
  7. While the total number of EEG samples is large (203,520), these are derived from only 60 subjects (30 per group). For deep learning models (LSTM, CNN), this is a very small cohort and raises concerns about potential overfitting and questions about the models' ability to generalize to the broader population.
  8. The stimuli are exclusively visual (RSVEP, SSVEP, emotional videos). PD is a motor disorder, and the study does not incorporate any motor tasks (e.g., finger tapping, gait analysis) or other cognitive tasks beyond emotional response. Incorporating such paradigms could capture more specific PD-related neural deficits and strengthen the findings.
  9. The results are presented as average accuracies per channel, but the paper lacks formal statistical tests to compare the performance of the four algorithms (SVM, CRC, LSTM, CNN) against each other. It is concluded that CRC and LSTM are best, but without statistical significance testing, this claim is somewhat anecdotal.
  10. The study excellently identifies optimal single channels but does not discuss the practical challenges of implementing a single-channel PD diagnostic device. A discussion on the signal-to-noise ratio challenges, the need for extremely precise electrode placement, and the validation required for such a minimal setup is missing.
  11. The method for ensuring the healthy control group had no underlying neurological conditions is not detailed. This is a critical step to avoid confounding results.

Author Response

We thank the reviewer for the valuable comments and suggestions on our manuscript. A detailed response to the review comments is mentioned in the attached file, which is also appended with the revised manuscript. 

Reviewer 2 Report

Comments and Suggestions for Authors

The submitted study introduces a cross-stimulation evaluation framework to analyze its impact on Parkinson’s disease detection algorithms, while also conducting a channel-wise analysis to identify the most discriminative brain regions for accurate diagnosis. This is indeed an interesting and potentially valuable contribution in the field of computational neuroscience and AI-based healthcare.

However, for the manuscript to achieve the necessary clarity and scientific rigor, a major structural revision is required. Specifically:

  1. Manuscript length: At present, the paper is overly lengthy, which dilutes focus on the primary objective. A more concise and streamlined narrative is strongly advised.

  2. Tables and figures: Several tables and figures do not add substantial value to the study. Non-essential or redundant elements should be removed, keeping only those that clearly support the results and conclusions.

  3. Focus on the main objective: The article must reinforce its focus on the stated aims, avoiding the inclusion of secondary or tangential content that does not directly contribute to the interpretation of outcomes.

  4. Methodology and data: While the methodology, sampling, and testing procedures are appropriate, they are currently obscured by excessive collateral information. A clearer and more targeted presentation would enhance the scientific strength of the findings.

In conclusion, this manuscript is based on a sound methodological approach and tackles an innovative and clinically relevant issue. Nevertheless, it requires structural and visual streamlining to maximize impact, readability, and scientific consistency. After such adjustments, it could be reconsidered for further evaluation.

Author Response

(The authors gave the same response as above.)

Reviewer 3 Report

Comments and Suggestions for Authors

A good effort, but the state did not definitively show a Mark difference in Parkinson’s disease versus normal controls. In addition there weren’t enough data on what’s normal compared to what you can find in Parkinson’s disease. Additionally, medication is taken can affect EEG such as benzodiazepines or GABA type trucks And it is important to know the type of drugs taken when the e.g. was done as Disque can significantly affect EEG.

I would revise the manuscript to include that these are limitations of this study and that it does not give enough evidence for the record relate between EEG and Parkinson’s disease also missing our unified disease, Parkinson’s disease rating skills or how did the patient get diagnosed? You would also need to put their motor scores to correlate with the EEG.

The study has significant limitations. It is an AI generated study Format which is good, but if you’re going to try to review Parkinson’s disease, you need to put some disease characteristics along with the correlation of findings in your visually potential or EEG.

I wasn’t able to follow the correlation with the visually Volk potential as that typically checks the intactness of the optic nerve track, which isn’t typically known to be affected in Parkinson’s disease so you would need to show studies that show this is affected in Parkinson’s disease before starting this study.

The study design needs to be redone, and most likely the findings will be challenged by many clinicians who read the study.

You may want to limited to the EEG.

Author Response

(The authors gave the same response as above.)

Reviewer 4 Report

Comments and Suggestions for Authors

Stimulus Evoked Brain Signals for Parkinson’s Detection: A Comprehensive Benchmark Performance Analysis on Cross Stimulation and Channel-Wise Experiments

I have read the manuscript with interest, and you can find my appraisal, suggestions, and concerns, section by section as follows:

Introduction: The introduction is well-written. However, I suggest adding more information about the sex that is more affected by PD. Similarly, PD is a very complex syndrome, and you need to be more specific in stating it in the introduction. Since you have focused on the motor issue in PD, I also suggest adding more information being more specific about the cognitive functions that are affected by dopamine dysfunction. In a similar way, dopamine is not the only stream affected in PD, and it needs to be added to the introduction, as well as the related references. Please add the required information. The description of the previous literature about the potential application of the EEG is ok. I also agree with your rationale about the fact that only recently, EEG was “discovered “ as a less expensive, flexible technique that can be used for several neuropsychiatric disorders. Despite this, and I agree with you, the motor issue, with involuntary movements in PD, could represent a source of artifacts in the EEG signal, acting in a non-predictable way. Moreover, EEG cap montage could stress patients, inducing a series of behavioral alterations. Please, take into account these considerations.

Since the specific topic and target of the journal, I do not want to underline the fact that the diagnostic process is not well-explained, since the diagnosis cannot be performed by only an “expert clinician”, but it is more complex.

The hypotheses and the purpose at the end of the section are well-explained, concise, and clear.

Methods: The section is well-written, allowing replication of the study. Please, add the source of the stimuli (YouTube or a specific film etc.). Moreover, please add the mean age of the two groups and specify if they were matched for sex, age, and education levels. I have appreciated the figures that are clear and informative.

The procedure that you applied is interesting and well-described. However, in 2.3.2, the regularization

Parameter needs to be explained better. The other parameters are clear.

Despite this, the remaining subsections are clear.

Results: The results are interesting, and SVM, compared to the other methods applied, showed its limitation in the accuracy, maybe depending on the nature of the signal. Interestingly, the horror clips confirmed previous results about the impact of negative emotions on brain processing. The results are interesting, since they allow to classify the two groups on the basis of high-order cognitive processing.

In the discussion, I suggest highlighting better the limitations of the study.

Author Response

(The authors gave the same response as above.)

Round 2

Reviewer 1 Report

Comments and Suggestions for Authors

The EEG, the RSVEP and the SSVEP may be a method for detection of some neurological disease, but not optimal for movement disorders like PD. In the real-world clinical practice, the problem is not to differentiate PD and HC, but to differentiate PD with a bunch of PD-like diseases, including PSP, MSA, etc..

Another example is, most PD patients present with tremor, and in EEG, people will only need to find the one with more motion artifacts in the time domain to locate PD patients in a mixed group of PD and HC. And there are symptoms other than tremor which will cause distinctive artifacts in EEG. The symptom dimension should be documented with UPDRS. 

Indeed, we can find EEG difference between PD and HC, but it's not a very clinically meaningful topic to use EEG for detecting PD from HC, let alone the data-driven method to implement the classification. 

In addition, the Unified Parkinson’s Disease Rating Scale were for PD severity rating but not diagnosis, so the diagnosis of PD in this study should also be questioned.

The medication state during EEG recording was not documented.

Author Response

(The authors gave the same response as above.)

Reviewer 2 Report

Comments and Suggestions for Authors

The authors have made the suggested changes.

Author Response

We sincerely thank the reviewers for their constructive feedback, which has helped us to strengthen the clarity, accuracy, and scientific rigor of our manuscript.

Reviewer 3 Report

Comments and Suggestions for Authors

I have reviewed the authors responses, and I’m still wanting to know if the authors put the type of medication the patient was on and acknowledge that carbidopa levodopa can indeed change beta waves and beta frequency .

in terms of coating literature to show changes in EEG patterns in Parkinson’s disease, there is also the limitation of some Parkinson’s disease can be parkinsonism, such as drug and juice parkinsonism are a typical parkinsonism. I did not see the author’s talking about this possibility and I would recommend placing it as a limitation.

One for the revision would be to put the limitation listed above such as medication use was not known and this patient may develop a typical Parkinson. I’m in the future that the study has limitations and in the future this data collection may be helpful..

Author Response

(The authors gave the same response as above.)

Reviewer 4 Report

Comments and Suggestions for Authors

The manuscript has been improved, and the authors have addressed my concerns satisfactorily. 

Author Response

(The authors gave the same response as above.)

Round 3

Reviewer 1 Report

Comments and Suggestions for Authors

Since the publication of previous UKPDS Brain Bank Diagnostic Criteria, knowledge has advanced, and concepts of the disease are shifting, changes have been made in the MDS version of diagnostic criteria early in 2015 and late in some others.

Also, the "participants were instructed to refrain from taking their regular medication on the day of recording" could not minimize pharmacological influences. This design depend on what kind of drug the patients need, but as far as I know, longer time for drug wash out is needed for any of these antiparkinson drugs .

Author Response

(The authors gave the same response as above.)

Reviewer 3 Report

Comments and Suggestions for Authors

Authors have responded to all concerns amd placed limitations 

Author Response

We thank the reviewer for their positive feedback and confirmation that all concerns have been addressed.